# Rho1 and Rgf1 establish a new actin-dependent signal to determine growth poles in yeast independently of microtubules and the Tea1–Tea4 complex

Patricia Garcia �ORCID *, Ruben Celador, Tomas Edreira, Yolanda Sanchez ORCID *

Instituto de Biología Funcional y Genómica (IBFG), CSIC/Universidad de Salamanca and Departamento de Microbiología y Genética, Universidad de Salamanca. C/ Zacarías González, Salamanca, Spain

* pgr@usal.es (PG); ysm@usal.es (YS)

**Data Availability Statement:** All relevant data are within the paper and its Supporting Information files.

## Abstract

Cellular asymmetry begins with the selection of a discrete point on the cell surface that triggers Rho-GTPases activation and localized assembly of the cytoskeleton to establish new growth zones. The cylindrical shape of fission yeast is organized by microtubules (MT) that deliver the landmark Tea1–Tea4 complex at the cell tips to define the growth poles. However, only a few *tea1Δ* cells mistaken the direction of growth, indicating that they manage to detect their growth sites. Here, we show that Rgf1 (Rho1-GEF) and Tea4 are components of the same complex and that Rgf1 activity toward Rho1 is required for strengthen Tea4 at the cell tips. Moreover, in cells lacking Tea1, selection of the correct growth site depends on Rgf1 and on a correctly polarized actin cytoskeleton, both necessary for Rho1 activation at the pole. We propose an actin-dependent mechanism driven by Rgf1–Rho1 that marks the poles independently of MTs and the Tea1–Tea4 complex.

## Introduction

Cell polarity is the primary mechanism for generating cellular asymmetry, which is critical for most cell and tissue functions such as development, cell migration, and differentiation in a wide variety of organisms including humans. It typically begins with a signal on the cell surface that triggers a cascade of molecular events that induce the localized assembly of cytoskeletal and signaling networks, which subsequently direct the formation of a new growth area [1,2]. Fission yeast has been used over the last decades as a simpler and more accessible model for studying this complex process [3]. Its cells have a cylindrical shape that is maintained throughout the entire cell cycle, changing in length but not in diameter. This phenomenon is achieved by restricting growth to the cell poles, a process that is still not well understood. Growth occurring at the cell ends has been mainly studied in the transition from monopolar to bipolar growth, termed New End Take-Off (NETO) [4–7]. NETO depends on specific polarity determinants, the kelch-repeat protein Tea1, the SH3 domain-containing protein Tea4 and the DYRK kinase, Pom1 among others [8–11]. In the absence of Tea1–Tea4 complex, cells grow

**Funding:** This work was supported by a grant PID2020-115111GB-I00 (Ministerio de Ciencia e Innovación, MICINN, Spain) to Y.S. The PhD fellowship (FPU program) from Ministerio de Ciencia e Innovación, MICINN, Spain supported RC. The Universidad de Salamanca postdoctoral fellowship (Programa II) supported PG. A contract obtained through the Consejería de Educación de la Junta de Castilla y León was granted to TE. The funders had no role in study design, data collection and analysis, decision to publish, or preparation of the manuscript.

**Competing interests:** The authors have declared that no competing interests exist

**Abbreviations:** BSA, bovine serum albumin; GS, Glutathione Sepharose; MT, microtubule; NETO, New End Take-Off; PBS, phosphate-buffered saline; PA, phosphatidic acid; PH, pleckstrin homology; PM, plasma membrane; SAP, stress-activated pathway; SDS, sodium dodecyl sulfate; SD, standard deviation; SRRF, super-resolution radial fluctuations.

monopolarly but maintain their cylindrical shape. However, under certain stresses, the cells often choose the wrong growth site, forming bulged and T-shaped cells [10,12].

Tea1 and Tea4 ride on growing microtubule (MT) plus ends to the cell tips, where they are released as discrete "dots" at the cortex, being Tea4 totally dependent on Tea1 for its location [12–15]. At poles, the prenylated protein Mod5 and the ERM (Ezrin-Radixin-Moesin) family protein Tea3 anchor Tea1 to the cell cortex [16,17]. Tea1 and Tea4 colocalize at the cell tips within sterol-rich membrane domains [18] to form clusters or nodes [19], and it is assumed that this association promotes the binding of other polarity factors in large protein complexes that organize polarized growth [4,10,20]. One of these proteins is the formin For3, whose association with polarity markers likely brings it into the proximity of activators, stimulating the formation of F-actin cables that will deliver growth cargo to the tip [9]. Nevertheless, in the absence of both MTs and actin cables fission yeast still polarize growth, although less efficiently [21].

Establishing polarized growth involves hundreds of proteins; however, a constant from yeast to humans involves local accumulation of active GTP-bound forms of Rho-family GTPases at the cell cortex [22–24]. While polarity is mostly associated with the functions of Rac and Cdc42 in mammals or Cdc42 in yeast, other GTPases such as RhoA or Rho1 (yeast) can also play a role in the development of polarity. Active RhoA is found at the leading edge as the edge advances in migrating cells, whereas Cdc42 and Rac1 are activated later [25]. In budding and fission yeast, Cdc42 displays the ability to polarize spontaneously [24,26–28]. However, the role of Rho1, the other essential GTPase, in polarized growth remains undefined. Depletion of Rho1 in growing cells induces shrinking and death via a kind of "apoptosis" that is accompanied by the disappearance of polymerized actin [29,30]. Rho1 activity is regulated by 3 GEFs, Rgf1, Rgf2, and Rgf3 that catalyze the exchange of GDP for GTP, rendering the GTPase in an active state [30–36]. The main activator of Rho1, Rgf1, regulates cell integrity through Rho1 by activating the β-glucan-synthase complex [30] and gene expression via the Pmk1 MAPK cell integrity-signaling pathway [30,37,38]. Moreover, Rgf1, like Tea1, Tea4 and other polarity factors, is required for the actin reorganization necessary to switch from monopolar to bipolar growth during NETO [30].

Here, we have studied this phenomenon to show that Rgf1 interacts with the cell end marker Tea4 and binds to the plasma membrane (PM) through its PH domain. Both PM-binding and Rho-GEF activity are required for stable accumulation of polarity markers at the cell poles. In addition, we have uncovered a new role for Rgf1 in restricting growth to the poles in the absence of polarity markers. Most *tea1Δ* cells maintain their cylindrical morphology unless subjected to stresses, suggesting that these cells detect the location of its poles by an unknown mechanism. Here, we show that this mechanism depends on the actin cytoskeleton and Rho1 activation by Rgf1. Therefore, we propose 2 parallel pathways to define the growth poles in fission yeast: the canonical one dependent on MTs and Tea1–Tea4 and another one dependent on actin and Rgf1–Rho1, both necessary to maintain a straight shape when the other is impaired.

## Results

### Rgf1 is required for proper localization of the Tea1–Tea4 complex

We have previously shown that *rgf1Δ* strain displays defects in bipolar growth, with approximately 80% of the cells showing monopolar pattern of growth compared to approximately 25% of the wild-type cells [30]. This growth defect has been described in mutants affected in polarity factors such as Bud6, Tea1, or For3 [39–41], suggesting that similarly to these proteins, Rgf1 triggers NETO, and thus *rgf1Δ* cells may have problems when choosing the right end of

growth. This prompted us to examine the localization of the landmark proteins Tea4 and Tea1 in cells lacking Rgf1. While Tea4-GFP was concentrated at both cell tips in wild-type cells [9,11], in *rgf1Δ* cells the signal detected was visibly diminished. Accordingly, a reduction in Tea1 and Tea4 localization in *rgf1Δ* cells has been described previously in a wide protein localization dependency study [42]. We compared the fluorescence intensity of Tea4-GFP in wild-type and *rgf1Δ* cells in the same preparation. We incubated *tea4-GFP rgf1⁺ sad1-dsred* (spindle pole body marker) cells and *tea4-GFP rgf1Δ* cells separately, and then mixed and imaged them at the same time (Fig 1A). In *rgf1Δ* cells, the Tea4-GFP signal was dispersed in small dots that spread out at the ends. The average fluorescence of Tea4-GFP dots at the tips of *rgf1Δ* cells was approximately half of that seen at the tips of the wild-type cells. Because the *rgf1Δ* cells grew in a monopolar fashion, we examined whether the Tea4 dots were more prominent at one end or whether they were similarly distributed at both ends. We observed that both wild-type and *rgf1Δ* cells accumulated Tea4 more at the non-growing end (revealed by calcofluor staining). However, compared to wild-type cells, the average fluorescence intensity of Tea4 in *rgf1Δ* cells was significantly reduced at both ends, although the decrease was greater at the growing pole (Fig 1B). This reduction is characteristic of *rgf1Δ* cells. In contrast, when performing the same experiment in *cdc10-129* cells, which exhibit a high percentage of monopolar cells at 37˚C (approximately 90%), we observed no appreciable differences in Tea4 fluorescence intensity at either end compared to wild-type cells (S1A Fig). Thus, Rgf1 is more important to localize Tea4 to the growing tip, the one where Rgf1 concentrates in wild-type monopolar cells (S1B Fig).

We also evaluated whether the localization of Tea1, which functions in a complex with Tea4 [9], is affected in the Rgf1 mutant. The wide co-localization between Tea1 and Tea4 in the absence of Rgf1 indicates that both proteins displayed similar localization defects (Figs 1C and S1C), and that similarly to wild-type cells, in *rgf1Δ* cells Tea1 and Tea4 were still bound and formed a complex. In *tea4Δ* cells, Tea1 is concentrated at the non-growing cell tip [9]. Given that in *rgf1Δ* cells, Tea1 also localized mainly to the non-growing end (S1D Fig), our results suggest that the Tea1 mislocalization could be a consequence of Tea4 mislocalization. We noticed that *rgf1Δ* cells showed a greater number of Tea4 discrete dots in the middle zone of the cell (Figs 1A, 1B and S1E). Examination of Tea4-GFP in wild-type and *rgf1Δ* cells with α–tubulin labeled in red (mCherry-Atb2) showed co-localization of Tea4 cytoplasmic dots with MTs (Fig 1D). Thus, in the *rgf1Δ* cells, a larger free cytoplasmic pool of Tea4, which is not properly sequestered at the poles, could now be available to be redirected to the cell cortex by MTs.

## Rgf1 functions to integrate Tea4 in big clusters at the cell tip

Next, we determined whether Tea4 was accurately delivered to the cell cortex in *rgf1Δ* cells by taking time-lapse images every 15 s. We did not observe appreciable differences in the delivery of Tea4 to the cell cortex between *rgf1⁺* and *rgf1Δ* cells. However, the Tea4-GFP signal failed to remain in the pole in *rgf1Δ* cells (S1 and S2 Movies). This can be better observed in the kymographs shown in Fig 1E, where the fluorescence of Tea4 vanished from the cell cortex of the *rgf1Δ* cells in a few seconds, whereas it remained stable in control cells.

Polarity factors such as Tea1 and Tea4 localize to the cellular cortex in discrete clusters [19], which are not easily observable when taking conventional lateral cell images. To better perceive the formation of Tea4 nodes at the cell poles, we used super-resolution radial fluctuations (SRRF) microscopy for "head-on" imaging of cell tips. We performed short time-lapse experiments (3 min) at the cellular tip cortex on "head-on" wild-type and *rgf1Δ* cells (tagged with Tea4-GFP and mCherry-Atb2). In wild-type cells, Tea4 nodes deposited by MTs remained

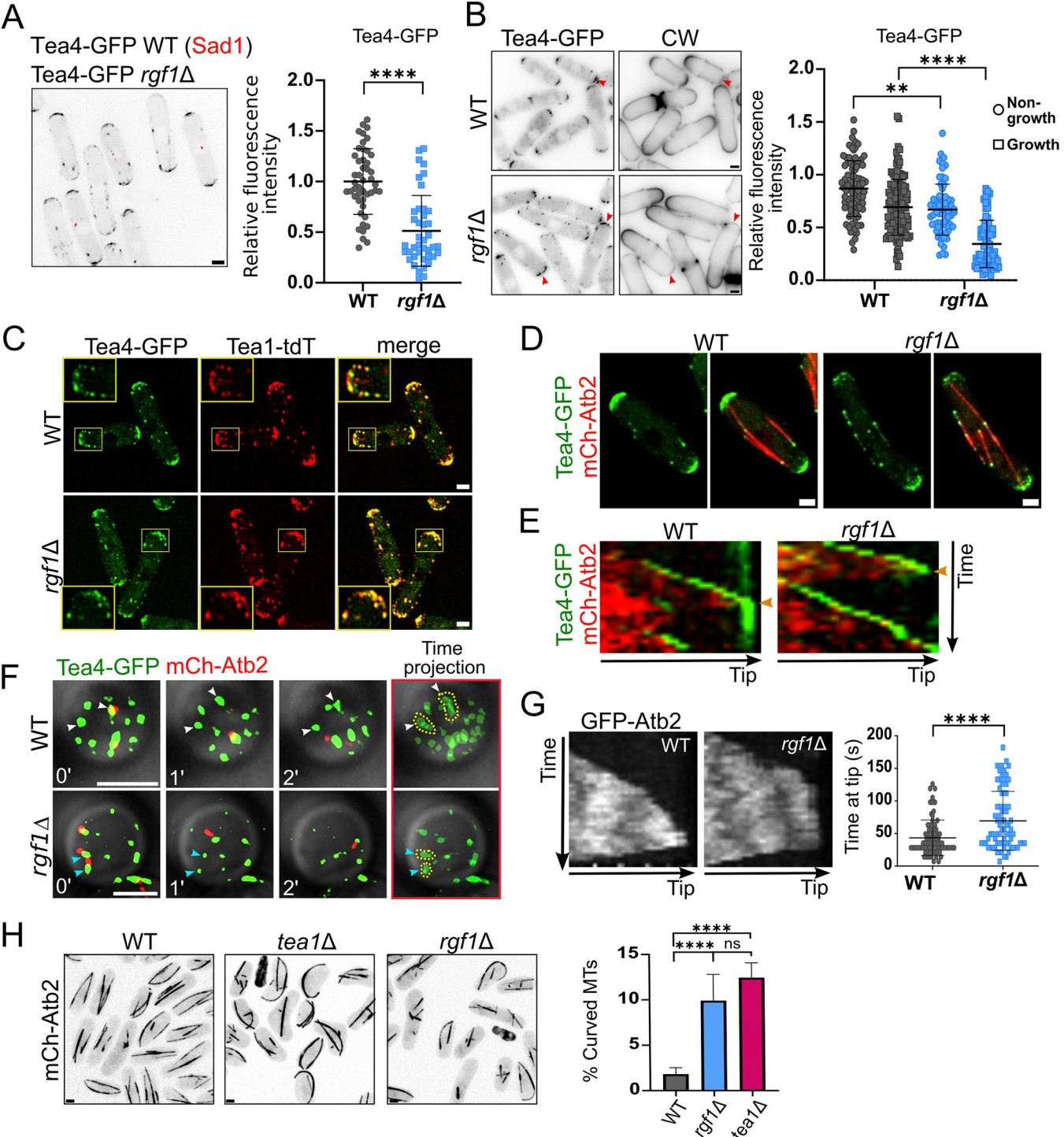

**Fig 1. Rgf1 is required for proper localization of the Tea1–Tea4 complex at the cell tip.** (A) Cells expressing *tea4-GFP rgf1⁺ sad1-dsred* and *tea4-GFP rgf1Δ* were grown in YES liquid medium separately and then mixed and imaged in the same preparation. The maximum-intensity projection of 6 deconvolved Z-slides (0.5 μm step-size) is shown. The graphic represents the mean ± SD of the relative fluorescence intensity of Tea4-GFP measured at the cell tips in the WT (*n* = 47) and *rgf1Δ* (*n* = 41) cells. WT levels were used for normalization. (B) Calcofluor white (CW, 20 μg/ml) staining and GFP fluorescence of cells expressing *tea4-GFP rgf1⁺* and *tea4-GFP rgf1Δ*. The maximum-intensity projection of 6 Z-slides (0.5 μm step-size) of Tea4-GFP fluorescence is shown. The red arrowheads indicate non-growing poles. The graphic represents the mean ± SD of the relative fluorescence intensity of Tea4-GFP measured at the growing end and the non-growing end of the WT and *rgf1Δ* (*n* >70) cells. Calcofluor staining, which marks sites of cell growth, was used to differentiate growth from non-growth poles (right). (C) Representative images of Tea4-GFP (green) and Tea1-tdTomato (red) localization in the WT and *rgf1Δ* cells. The maximum-intensity projection of 6 deconvolved Z-slides (0.5 μm step-size) is shown. (D) Representative images of the indicated cells expressing *tea4-GFP* (green) and *mCherry-*

*atb2* (red). The maximum-intensity projection of 6 Z-slides (0.5 μm step-size) is shown. (E) Kymographs of time-lapse fluorescence movies of *tea4-GFP* and *mCherry-atb2* expressed in WT or *rgf1Δ* strains. The maximum-intensity projection of 7 Z-slides (0.6 μm step-size) of images taken every 15 s was used to draw a line along an MT from the middle of the cell to the tip. The orange arrowheads indicate the moment of MT retraction. (F) SRRF images of Tea4-GFP (green) and mCherry-Atb2 (red) in "head-on" cell tips of the indicated strains. One focal plane image was taken every minute. The time projection of the 3 images at different time points is shown to follow Tea4 cluster movement (right). Note that in the WT strain Tea4 nodes remain stable (white arrowheads) for longer than in the *rgf1Δ* mutant (blue arrowheads). (G) Kymographs of time-lapse fluorescence movies of GFP-Atb2 producing in the WT and *rgf1Δ* cells. The graph shows the mean ± SD of the time during which the MT is touching the pole in both strains (*n* = 75). (H) Representative images of the indicated cells producing mCherry-Atb2 growing for 4 h at 37˚C. The maximum-intensity projection of 6 Z-slides (0.5 μm step-size) is shown. The graph shows the mean ± SD of the percentage of curved MTs found in the indicated strains (*n* > 500). Statistical significance was calculated using two-tailed unpaired Student's *t* test. ****$P < 0.0001$; **$P < 0.01$; ns = nonsignificant. Scale bar, 2 μm. The data underlying the graphs shown in the figure can be found in S1 Data. MT, microtubule; SD, standard deviation; SRRF, super-resolution radial fluctuations; WT, wild type.

stable for an interval of time even after MT catastrophe (Fig 1F, white arrows). Moreover, clusters of Tea4 not associated with MTs could be observed stable throughout the time-lapse and were of the same size as those associated with MTs. However, in *rgf1Δ* cells, Tea4 dots of a similar size to those observed in the wild-type cells appeared exclusively while they were associated with MTs (Fig 1F, blue arrows). Once the MT retracted, the Tea4 cluster became gradually smaller until it eventually disappeared. These observations confirmed the results obtained with conventional microscopy methods, suggesting that Rgf1 was not necessary for the delivery of Tea4 to the cell poles, but it is required for its stable maintenance once it was released there. In addition, we ruled out a defect in the stability of the Tea4 protein in the *rgf1Δ* mutant because the half-life of the protein in the wild-type and *rgf1Δ* cultures treated with cycloheximide was comparable (S1F Fig).

During the course of these experiments, we noticed that some MTs curled around the tips in the *rgf1Δ* cells. We confirmed that the MT dynamics was not affected because the polymerization and depolymerization rates were similar in the wild-type and *rgf1Δ* cells, with a slight increase in the polymerization rate in the mutant (S1G Fig). However, the mean time that the MT stayed at the tip was approximately 43 s in the *rgf1+* cells but approximately 70 s in the *rgf1Δ* cells (Fig 1G). Therefore, once an MT reached the cortex in the *rgf1Δ* cells, it remained there longer than in the wild-type cells. Curved MTs have already been described in *tea1Δ* cells growing at a high temperature [10]. We incubated the *rgf1Δ* and *tea1Δ* mutants at 37˚C for 4 h and observed MT organization under these conditions (Fig 1H). In the *rgf1Δ* mutant, approximately 10% of the cells possessed at least 1 MT curled around the end, similarly to the ~12% found in the *tea1Δ* mutant, while this type of curly MT was rarely observed in the wild-type (1.7%). It is possible that a limited amount of Tea4–Tea1 at the growing pole underlies the curly phenotype seen in the absence of Rgf1.

## Rgf1 cooperates with Mod5 in Tea4 anchoring to the cellular poles

It has been shown that in cells lacking Mod5, Tea1 and Tea4 fail to accumulate to wild-type levels at the cell tips [9,16,43]. Given that the *rgf1Δ* cells showed a similar defect, we analyzed the localization of Tea4 and Tea1 in the double mutant *rgf1Δmod5Δ* compared with the single mutants. In the *mod5Δ* and *rgf1Δ* cells, most of the Tea4-GFP dots were associated with MTs, although was still some signal mainly at one of the poles (Fig 2A, yellow arrows). However, in the *rgf1Δmod5Δ* double mutant, the Tea4 signal was depleted at both poles (Fig 2A, orange arrows). The Tea1 signal at the poles exhibited a pattern similar to that of Tea4 in both single and double mutants (S2A Fig), suggesting that Mod5 and Rgf1 share a function in Tea1–Tea4 anchoring.

Because the localization of Tea4 is entirely dependent on Tea1 and the localization of Tea1 is partially dependent on Tea4 [9], we wondered whether the phenotype of the *rgf1Δmod5Δ* cells (which mislocalize Tea4) resemble the *tea1Δ* phenotype. To this end, we scored the

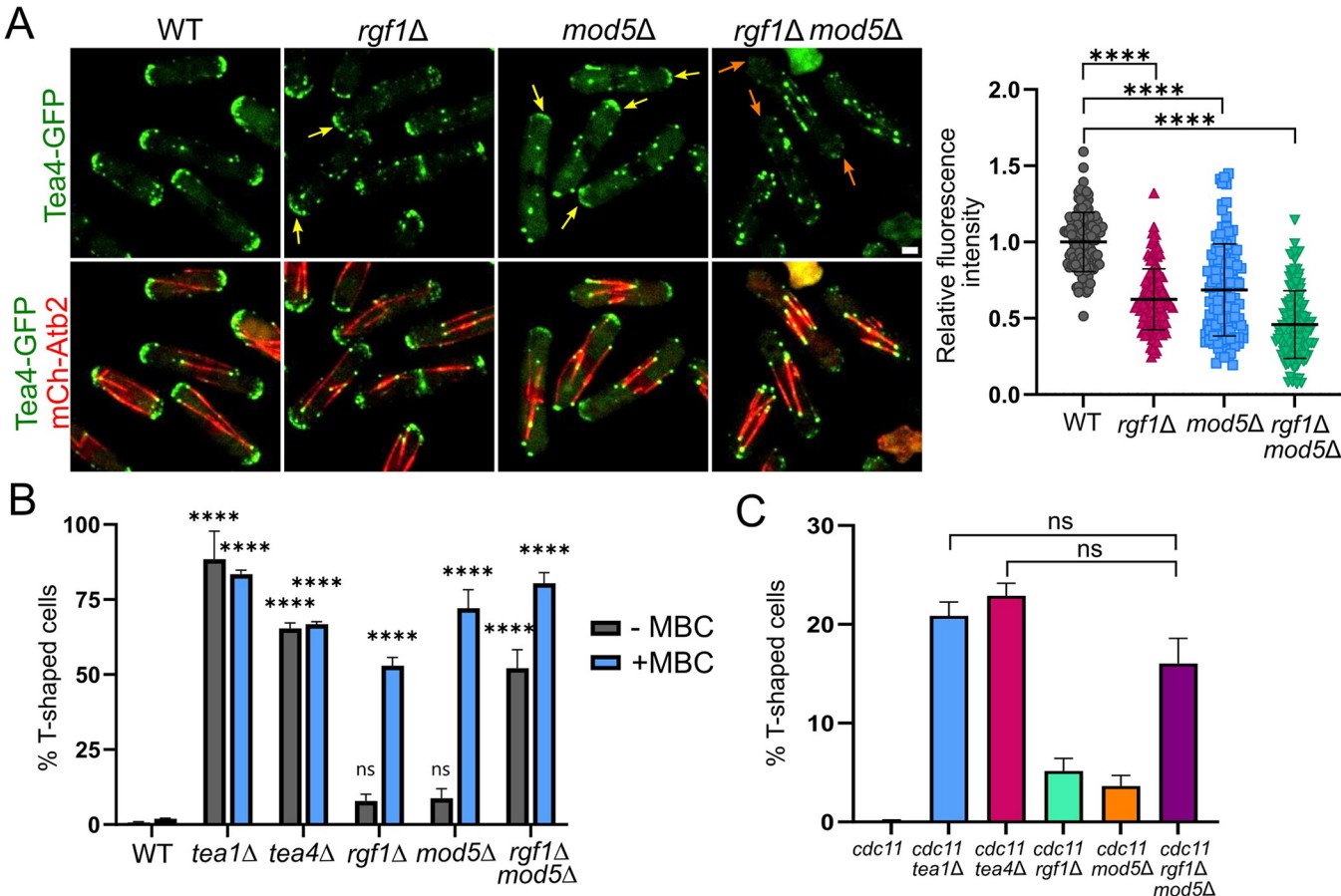

**Fig 2. Rgf1 cooperates with Mod5 in Tea4 anchoring to the cellular poles.** (A) Representative images of the indicated strains producing Tea4-GFP (green) and mCherry-Atb2 (red). The maximum-intensity projection of 6 Z-slides (0.5 μm step-size) is shown. Scale bar, 2 μm. The graphic represents the mean ± SD of the relative fluorescence intensity of Tea4-GFP measured at the cell tips in the WT, *rgf1Δ*, *mod5Δ*, and *rgf1Δ mod5Δ* cells (*n* > 120). WT levels were used for normalization. Statistical significance was calculated using a two-tailed unpaired Student's *t* test. (B) The percentage of cells forming branches in the indicated strains. The cells were grown to the stationary phase for 3 days at 28˚C, and then were treated with DMSO (-MBC) or MBC (+MBC; 50 μg/ml) in fresh medium for 3 h. The mean ± SD of >200 cells from 3 independent experiments is shown. Statistical significance of each strain compared to the WT was calculated using a two-way ANOVA test. (C) The percentage of T-shaped cells in the indicated strains after incubation for 4 h at 36˚C in YES liquid medium. The experiment was repeated at least 5 times. Statistical significance was calculated using a two-tailed unpaired Student's *t* test. ****$P$ < 0.0001; ns = nonsignificant. The data underlying the graphs shown in the figure can be found in S1 Data. SD, standard deviation; WT, wild type.

percentage of T-shaped cells in polarity re-establishment assays. In these assays, we first grew cells to the stationary phase for 3 days, and then diluted them in fresh medium for 3 h. This treatment increases the penetrance of polarity mutant phenotypes, inducing T-shapes and bulges and was performed either in the presence or in the absence of the MT inhibitor *methyl-2-benzimidazole carbamate* (*MBC*), to prevent or allow the continuous delivery of Tea1–Tea4 to the poles by MTs [12,16]. Consistent with the "poor" localization of Tea4 in the cell cortex, the *rgf1Δ* and *mod5Δ* cells showed polarity defects when treated with MBC (52% and 73% in *rgf1Δ* and *mod5Δ* cells, respectively) and marginal defects in the absence of MBC. In contrast, approximately 83% of the *tea1Δ* and 63% of the *tea4Δ* cells displayed the characteristic T-shaped pattern, with or without MBC treatment (Figs 2B and S2B) [16]. Interestingly, the *rgf1Δmod5Δ* cells behaved similar to *tea* mutants, reaching >50% of T-shaped cells even in the presence of MT (-MBC) (Figs 2B and S2B). In addition, we combined the *rgf1Δmod5Δ* deletion with the temperature sensitive (ts) septation mutant *cdc11-119*. At the restrictive temperature, *cdc11-119* cells show a defect in cytokinesis, but the nuclear and growth cycles continue and

cells grow at both ends after each mitosis. Presumably, after each mitosis, *cdc11-119* mutant cells must decide where to reinitiate growth. In the *cdc11-119 tea1Δ* and *cdc11-119 tea4Δ* double mutants, these events are often aberrant, leading to the formation of highly branched or T-shaped multi nuclei cells [10,44]. Only ~5% of the *cdc11rgf1Δ* and *cdc11mod5Δ* cells were T-shaped after incubation for 4 h at 36°C (Figs 2C and S2C). However, the *cdc11rgf1Δmod5Δ* triple mutant showed a similar percentage of T-shaped cells (approximately 17%) as the *cdc11tea1Δ* and *cdc11tea4Δ* double mutants (approximately 21% and approximately 24%, respectively). These results indicate that Rgf1 and Mod5 collaborate to position the polarity markers at the cell tips to prevent mislocalization of growth machinery in successive cell cycles.

## Rgf1 interacts with the cell-end marker Tea4 and binds to phosphatidylinositol-4-phosphate through its PH domain

Tea1 and Tea4 reside in large protein complexes [10,20]. We used different approaches to determine whether Rgf1 acts locally to retain Tea4 at the cell tips. First, we examined the in vivo localization of endogenous Tea4-GFP together with Rgf1-tdTomato at the cell poles by SRRF microscopy for "head-on" imaging of cell tips. In interphase cells, a subset of Rgf1 dots colocalized with Tea4 dots at cell tips, indicating a close proximity with each other (Figs 3A and S3A). Subsequently, we tested for the coprecipitation of these 2 proteins from yeast extracts by using epitope-tagged strains. Indeed, endogenously expressed GFP-tagged Tea4 led to the co-purification of HA-tagged Rgf1 (Fig 3B). Tea1 forms a stable complex with Tea4 [9]; however, we could not detect the interaction between Tea1 and Rgf1 (S3B Fig). To validate these associations, we purified GST-Rgf1 from *Escherichia coli* and conjugated it with Glutathione Sepharose (GS) beads; then, we used those beads in a pull-down assay to trap Tea4-GFP or Tea1-GFP from *S. pombe* protein extracts. We detected binding of Tea4 when using Rgf1-GS beads but not with GS beads alone (Fig 3C). In addition, we could observe a slight precipitation of Tea1, which might be the result of Tea4 interaction with Tea1. We confirmed the biochemical interaction in a two-hybrid assay. Consistently, Rgf1 could interact with Tea4 but not with Tea1 (Fig 3D). Taken together, these results indicate that Rgf1 associates with the Tea1–Tea4 complex through its binding with Tea4.

Next, we evaluated whether Rgf1 acts as a linker between Tea4 and the PM in the anchoring process. Rgf1 is a large (approximately 150 KDa) multi-domain protein, including a pleckstrin homology (PH) domain [36]. PH domains could act as a "membrane-targeting device" by anchoring GEFs to phosphoinositides and directing them towards their partner GTPases on the cellular cortex [45,46]. To test the ability of Rgf1 to bind different membrane lipids, we fused the Rgf1 protein to GST (without its carboxi-terminal CNH domain to purify it more easily) and purified it from bacteria. We used membrane arrays spotted with different kind of lipids to detect binding of GST-Rgf1 to lipids. As shown in Fig 3E, recombinant-purified GST-Rgf1 preferentially bound to phosphatidic acid (PA) and cardiolipin and also bound to phosphatidylinositol 4-phosphate [PI(4)P], and 3-sulfogalactosylceramide more subtly. GST-Rgf1ΔPH (additionally lacking the PH domain) could not interact with PI(4)P, but it behaved like the wild-type protein in terms of its binding to the other lipids. This result contrasts with that described for the budding yeast Rgf1 homolog, Rom2, whose PH domain specifically binds to phosphatidylinositol 4,5-bisphosphate [PI(4,5)P$_2$] [47]. Thus, in *S. pombe* Rgf1 might bind to the PM through the interaction between its PH domain with the phospholipid PI(4)P. Consistently, the PH domain was required for proper anchoring of Rgf1 to the cellular cortex. A GFP-tagged Rgf1ΔPH mutant showed large defects in its localization at both the poles and the septum (Fig 3F). The fluorescence detected at the *rgf1ΔPH-GFP* cell ends was approximately 25% of that exhibited by the full-length protein Rgf1-GFP (Fig 3F, right).

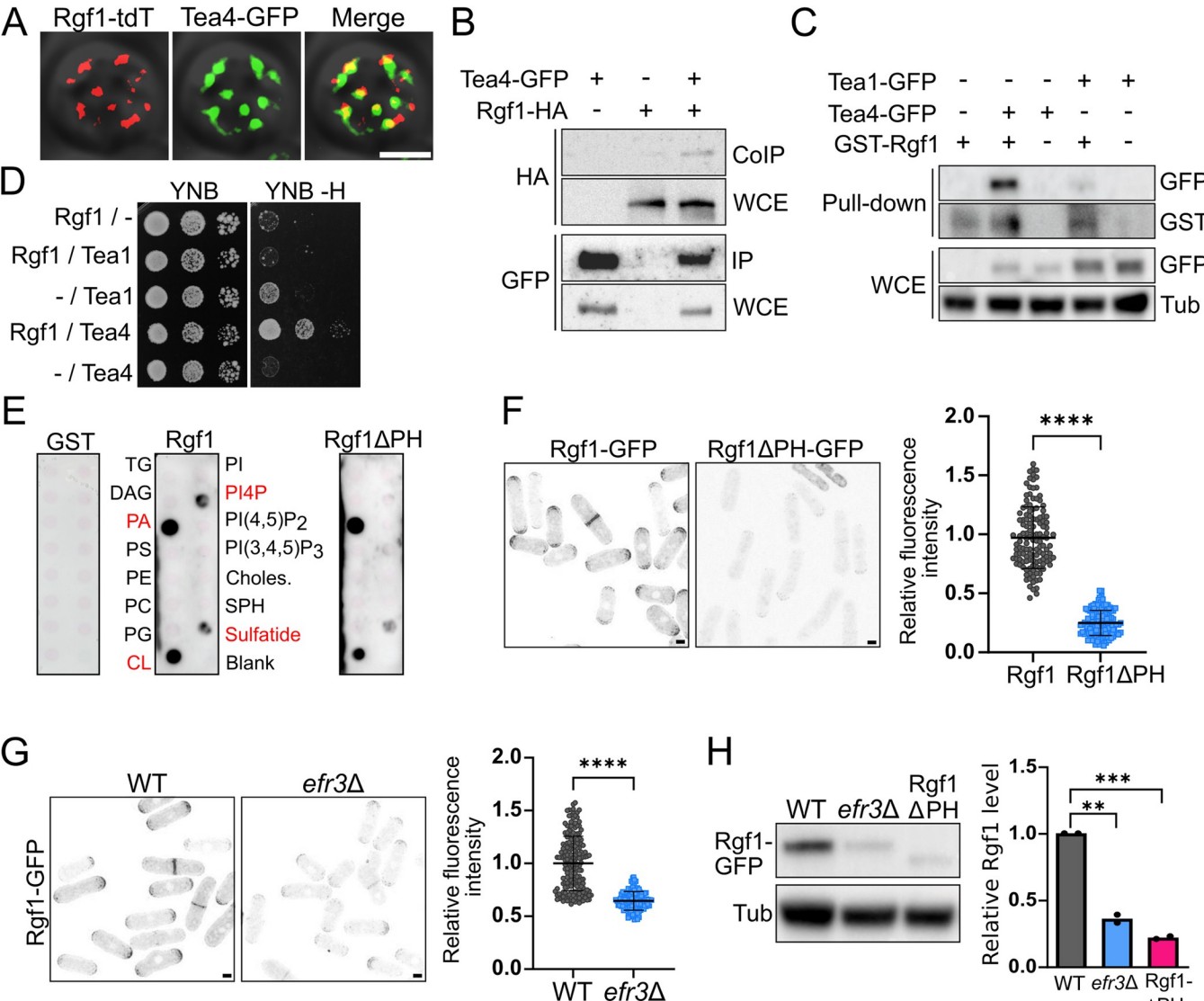

**Fig 3. Rgf1 interacts with the cell end marker Tea4 and binds to phosphatidylinositol-4-phosphate through its PH domain.** (A) Colocalization of Rgf1 and Tea4. SRRF images in "head-on" cell tips of the wild-type producing Tea4-GFP endogenously (green) and Rgf1-tdTomato from a plasmid under the control of its own promoter (red). (B) Coprecipitation of Rgf1 and Tea4. Cell extracts from cells producing Tea4-GFP, Rgf1-HA, or Tea4-GFP and Rgf1-HA were precipitated with GFP-trap beads and blotted with anti-HA or anti-GFP antibodies (co-immunoprecipitation and immunoprecipitation). Western blot was performed on total extracts to visualize total Tea4-GFP and Rgf1-HA levels (whole cell extracts). (C) Cells expressing Tea1-GFP or Tea4-GFP were pulled down from cell extracts with GST-Rgf1 purified from *E. coli* bound to GS-beads or with GS-beads alone, and blotted with anti-GST or anti-GFP antibodies (Pull-down). Total Tea1-GFP or Tea4-GFP levels (WCE) were visualized by western blot; tubulin was used as a loading control. (D) Two-hybrid analysis of the interaction between Tea1 (pGR135) and Tea4 (pGR106) with Rgf1 (pRZ97). The interaction was assessed on YNB plates without histidine (YNB -H), including Rgf1/pGAD (Rgf1/-), Tea1/pGBK (-/Tea1), and Tea4/pGBK (-/Tea4) as controls. (E) Protein-lipid overlay assay. Membrane lipid strips were overlaid with 1 μg/ml of the purified GST, GST-Rgf1, and GST-Rgf1ΔPH, respectively, and the interaction was detected with an anti-GST antibody. Lipids to which GST-Rgf1 showed a significant association are shown in red. Note that the interaction with PI4P disappears with GST-Rgf1ΔPH. (F) Representative images of cells producing Rgf1-GFP or Rgf1ΔPH-GFP. The maximum-intensity projection of 6 Z-slides (0.5 μm step-size) is shown. The graphic represents the mean ± SD of the relative fluorescence intensity measured at the cellular tips of Rgf1-GFP and Rgf1ΔPH-GFP (*n* > 120). (G) Representative images of cells producing Rgf1-GFP in WT or *efr3Δ* mutant. The maximum-intensity projection of 6 Z-slides (0.5 μm step-size) is shown. The graphic represents the mean ± SD of the relative fluorescence intensity measured at the cellular tips of Rgf1-GFP (*n* > 120). (H) Protein extracts from cell producing Rgf1-GFP in the WT or *efr3Δ* mutant and Rgf1ΔPH-GFP were analyzed by western blot with an anti-GFP antibody to visualize Rgf1 levels. An anti-tubulin antibody was used as a loading control. The graphic represents the mean ± SD of the relative Rgf1 proteins levels from 2 independent experiments. Statistical significance was calculated using a two-tailed unpaired Student's *t* test. ****$P < 0.0001$, ***$P < 0.001$, **$P < 0.01$. Scale bar, 2 μm. The data underlying the graphs shown in the figure can be found in S1 Data. GS, Glutathione Sepharose; PH, pleckstrin homology; SD, standard deviation; SRRF, super-resolution radial fluctuations.

It has been previously described that mutants in phosphatidylinositol kinases, which present an imbalance in the levels of PI(4)P and PI(4,5)$P_2$, show aberrant localization of Rgf1 in the PM [48,49]. Moreover, cells lacking $efr3^+$, a PM scaffold for the PI(4)P kinase Stt4, show an important reduction of Rgf1 at the cell division site [49]. We found that the localization of Rgf1 at the cell poles was compromised in the $efr3\Delta$ cells (Fig 3G). In addition, we detected a 3- to 4-fold decrease in the protein level of Rgf1, both without the PH domain or in $efr3\Delta$ cells, when Rgf1 was not properly bound to the PM (Fig 3H). Altogether, these results support a role for PM phospholipids in the anchoring of Rgf1 to the cellular cortex and in the maintenance of Rgf1 protein level.

## Tea4 accumulation at the cell ends depends on Rgf1 anchoring to the PM and Rho1 activation

Next, we determined whether Tea4 is bound to the cell cortex in the $rgf1\Delta PH$-GFP mutant, which shows compromised Rgf1 localization. Cortical localization of Tea4 was greatly reduced (approximately 35%) in the mutant lacking the PH domain compared with the wild-type ($P < 0.0001$) (Figs 4A and S4A). GTPase activation by its GEFs usually takes place when both the GTPase and the GEF are close to the PM; thus, a low level of Rgf1$\Delta$PH-GFP at the PM could promote inefficient activation of the Rho1 GTPase. To examine whether this is the case, we analyzed the in vivo amount of GTP-Rho1 (active-Rho1) in the $rgf1\Delta PH$ cells in a pull-down assay with GST-C21RBD, the rhotekin-binding domain (previously purified from bacteria) [30]. We found that the level of active-Rho1 detected in the $rgf1\Delta PH$ cells was similar to that seen in the $rgf1\Delta$ cells, and much less than the amount detected in control cells (Fig 4B). However, in the $rgf1\Delta PH$-GFP mutant the ability to bind and anchor Tea4 to the cortex and to activate Rho1 could be reduced due to a significant drop in the protein level compared with the Rgf1-GFP protein (Fig 3H). To determine if the instability of the $rgf1\Delta PH$ protein is behind the defects observed in the mutant, we replaced the $rgf1$ promoter with that of actin ($act1$), achieving levels of the truncated protein similar to those of the wild type (S4B Fig). Even so, pact-$rgf1\Delta PH$-GFP does not localize correctly at the cell poles and also fails to localize Tea4 or activate Rho1 efficiently (S4C–S4E Fig), indicating that the defects in the $rgf1\Delta PH$ mutant are not due to the low amount of protein but rather the absence of the PH domain. Both, the localization of Rgf1 to the PM and the activation of Rho1 were impaired in the $rgf1\Delta PH$ mutant; hence, we could not determine which one is behind Tea4 mislocalization. To distinguish between these 2 possibilities, we utilized the $rgf1$-$\Delta PTTR$ mutant expressing a protein without 4 amino acids in the RhoGEF domain. This mutant displays significantly reduced GEF activity toward Rho1 [38] while the protein remained attached to the growing end (S4F Fig). Tea4-GFP was also mislocalized in the $rgf1$-$\Delta PTTR$ mutant (Figs 4A and S4A), indicating that the stable association of Tea4 to the membrane is dependent on Rgf1 GEF activity.

We wondered whether the Rgf1-$\Delta PH$ and Rgf1-$\Delta PTTR$ mutant proteins retained the ability to bind Tea4 in vitro. Both proteins proficiently bound Tea4 in an in vitro pull-down assay (Fig 4C and 4D). In vivo, we analyzed the percentage of T-shaped cells after re-entry from the stationary phase to fresh medium. With MBC treatment, the percentage of T-shaped cells was approximately 50% for the $rgf1\Delta$ mutant; this percentage was similar for the $rgf1$-$\Delta PH$ (approximately 55%), with its own promoter or that of actin (Figs 4E and S4G) and $rgf1$-$\Delta PTTR$ (approximately 35%) mutants and much higher than for the wild-type strain (approximately 5%) (Fig 4E). Taken together, these results indicate that the localization of Rgf1 to the PM and its ability to activate Rho1 are closely linked; both functions are required to maintain Tea4 at the cell tips and to preserve the growth pattern after refeeding. The elimination of the PH domain and mutation of the catalytic site simultaneously has no additive effect, demonstrating

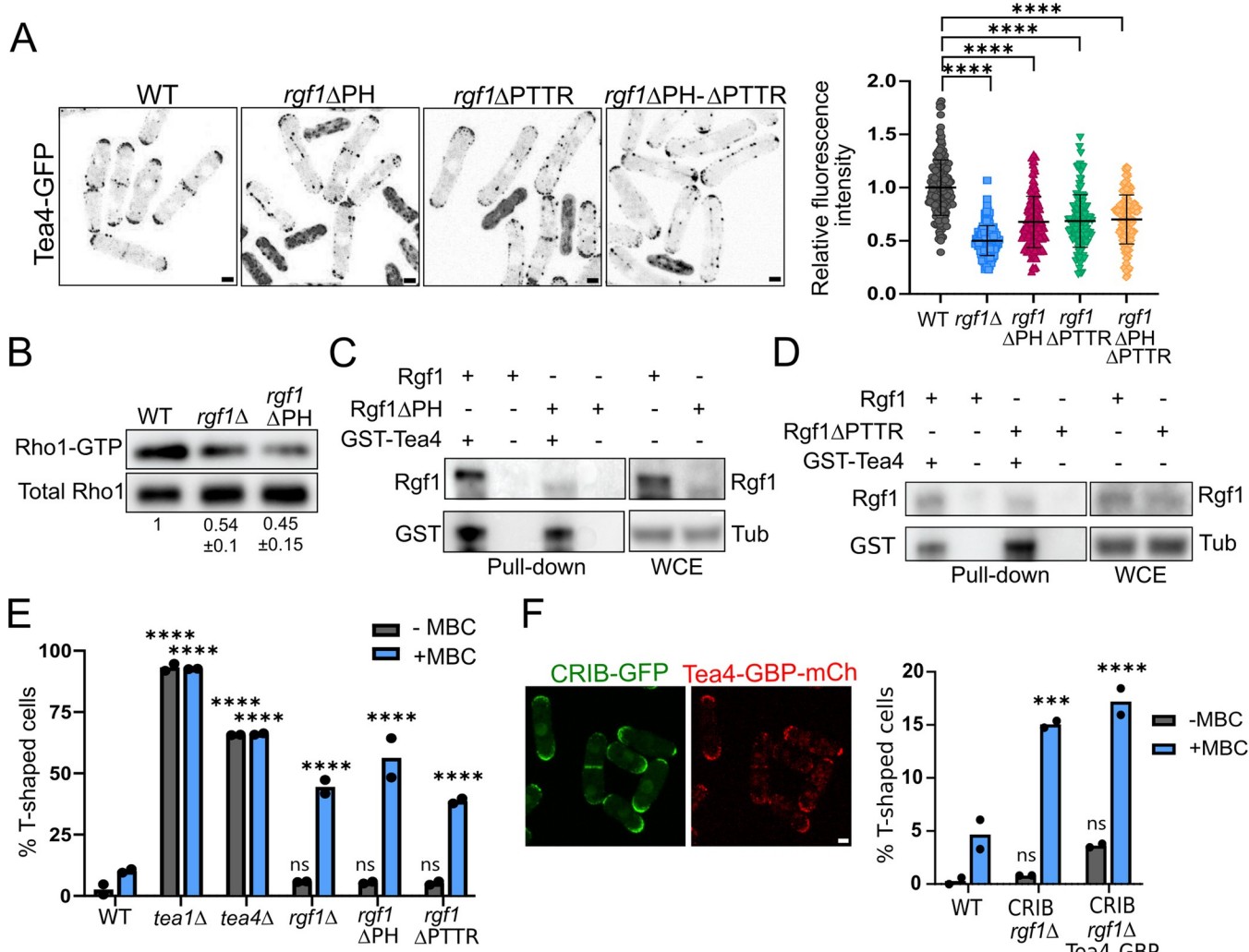

**Fig 4. Tea4 accumulation at the cell ends depends on Rgf1 anchoring to the PM and Rho1 activation.** (A) Representative images of Tea4-GFP localization in the WT, *rgf1Δ*, *rgf1ΔPH*, and *rgf1ΔPTTR* and *rgf1ΔPH-ΔPTTR* cells. The maximum-intensity projection of 6 Z-slides (0.5 μm step-size) of Tea4-GFP fluorescence is shown. Scale bar, 2 μm. The graphic represents the mean ± SD of the relative fluorescence intensity of Tea4-GFP (*n* > 130) measured at the cellular tips. Statistical significance was calculated using a two-tailed unpaired Student's *t* test. (B) Extracts from cells producing Rho1-HA (pREP4X-Rho1-HA) in the WT, *rgf1Δ*, and *rgf1ΔPH* cells were pulled down with GST-C21RBD and blotted against anti-HA antibody (Rho1-GTP). Total Rho1-HA was visualized by western blot (WCE). The relative units indicate the fold-differences in Rho1 levels in the mutants compared with the WT strain, with an assigned value of 1 (bottom) from 3 independent experiments. (C) Extracts from cells producing Rgf1-GFP or Rgf1ΔPH-GFP were pulled down with GST-Tea4 purified from *E. coli* bound to GS-beads or with GS-beads alone and blotted against anti-GST or anti-GFP antibodies (Pull-down). The total Rgf1 levels (WCE) were visualized by western blot; tubulin was used as a loading control. (D) Extracts from cells producing Rgf1-HA or Rgf1ΔPTTR-HA were pulled down with GST-Tea4 purified from *E. coli* bound to GS-beads or with GS-beads alone and blotted against anti-GST or anti-HA antibodies (Pull-down). Total Rgf1 levels (WCE) were visualized by western blot; tubulin was used as a loading control. (E) The percentage of cells WT, *tea1Δ*, *tea4Δ*, *rgf1Δ*, *rgf1ΔPH*, and *rgf1ΔPTTR* cells forming branches 3 h after release to growth after 3 days in stationary phase, in the absence and in the presence of MBC (50 μg/ml). Statistical significance of each strain compared to the wild type was calculated using a two-way ANOVA test. (F) Fluorescence images from log-phase *rgf1Δ* cells expressing CRIB-GFP (green) and Tea4-CBP-mCherry (red). The maximum-intensity projection of 6 Z-slides (0.5 μm step-size) is shown. Scale bar, 2 μm. The graph represents the percentage of cells forming branches after refeeding in the indicated strains. The cells were grown to the stationary phase for 3 days at 28°C, and then were treated with DMSO (-MBC) or MBC (+MBC; 50 μg/ml) in fresh medium for 3 h. The mean ± SD of >200 cells from 2 independent experiments is shown. Statistical significance was calculated using a two-way ANOVA test. ****$P < 0.0001$, ***$P < 0.001$, ns = nonsignificant. The data underlying the graphs shown in the figure can be found in S1 Data. GS, Glutathione Sepharose; PM, plasma membrane; SD, standard deviation; WT, wild type.

this link (Figs 4A, S4A and S4G). Furthermore, artificially targeting of Tea4 to the poles by adding a GBP domain to bind it to the CRIB-GFP protein does not restore the *rgf1Δ* defects observed in refeeding experiments (Fig 4F). In sum, these results indicate that Rho1 activation

is important for promoting the formation of a stable critical mass of Tea4 in the cell cortex that is essential to activate pole-restricted growth.

## Rgf1 is part of actin-dependent machinery that signals growth poles in the absence of the Tea1–Tea4 complex

As we have described above, *tea1Δ* cells experience an exacerbated loss of polarity when subjected to nutritional stress while keeping their cylindrical shape in normal conditions. We wondered why this occurs and how cells recognize their poles in the absence of the classical Tea markers. Since Tea4 localization is entirely dependent on Tea1 [9], we utilized *tea1Δ* mutant to eliminate both markers at the poles. It has been proposed that the high number of T-shaped cells observed in the *tea1Δ* mutant in refeeding experiments (Figs 2B and 4E) is due to a transient depolarization of the actin cytoskeleton [50]. To address this issue, we confirmed the disorganization of the actin cytoskeleton in wild-type cells grown for 3 days in liquid medium (Fig 5A). In addition, exposure to KCl, sorbitol, or heat, which promoted actin depolarization in wild-type cells also induced branching in the *tea1Δ* mutant (S5A and S5B Fig) [11,51]. We thought that if the mechanism that keeps the identity of the growth sites in the absence of Tea1 is lost because of transient actin depolarization, then treating *tea1Δ* cells with latrunculin A (LatA), which prevents the polymerization of filamentous actin, should increase the number of branched cells. As expected, 75% of the *tea1Δ* cells showed polarity defects during recovery from the LatA treatment, compared with approximately 2% of the wild-type and untreated *tea1Δ* cells (Fig 5B). Thus, a properly polarized actin cytoskeleton is required to position growth sites at opposite cell poles in the absence of Tea1–Tea4 markers.

Previously, we described that *rgf1Δ* cells show defects in actin reorganization during NETO [30]; however, other defects in actin patches and cables distribution were not taken account at that time. Now we analyzed the distribution of actin patches along the cell using confocal microscopy and LifeAct to mark actin. Our observations revealed a higher concentration of actin patches in the middle zone of the cell in the absence of *rgf1*, compared to the almost complete absence of these patches in the wild type (S5C Fig), suggesting that *rgf1Δ* cells also exhibit defects in the proper polarization of actin patches. Then, [30] we reasoned that the absence of Rgf1 in the *tea1Δ* background could uncover polarity defects that would otherwise remain undetected. This was indeed the case; 50% of the *tea1Δrgf1Δ* cells in unperturbed conditions were T-shaped compared with fewer than 5% of the *tea1Δ* and *tea1Δmod5Δ* cells (Fig 5C). These cells usually fail to choose the growth site after division. Frequently, the cell that inherits the non-growing end from the mother cell is unable to recognize or activate this old end and the growth machinery is organized at a new site on the lateral side (S3 Movie). Thus, in the absence of Rgf1 and polarity markers (Tea1–Tea4), stresses that induce actin depolarization were not necessary for branching to occur. Moreover, Rho1 activation was required to maintain polarity, evidenced by the high number of T-shaped cells observed in the *tea1Δrgf1Δ*PTTR mutant (Fig 5C). Therefore, in the absence of Tea1, Rgf1–Rho1 probably mark the poles through an actin-dependent mechanism.

To understand why the *tea1Δrgf1Δ* null cells behaved alike *tea1Δ* stressed cells, we studied the localization of Rgf1 under situations that depolarize the actin cytoskeleton. Rgf1-GFP localization to the cell tips was lost quickly in cells treated with LatA, but was unaffected in cells treated with MBC (S5D Fig). Thus, we reasoned that because actin is depolarized in quiescent cells, Rgf1-GFP should behave similarly. Accordingly, Rgf1 was missing from the cell periphery in stationary phase cells, while Tea1, Tea4, and Mod5 (which depend on MTs to reach the poles), remained polarized in the same experiment (Fig 5D) [10,11,16]. Moreover, Rgf1 disappeared from the cell tips under osmotic or heat stress but was observed in lateral

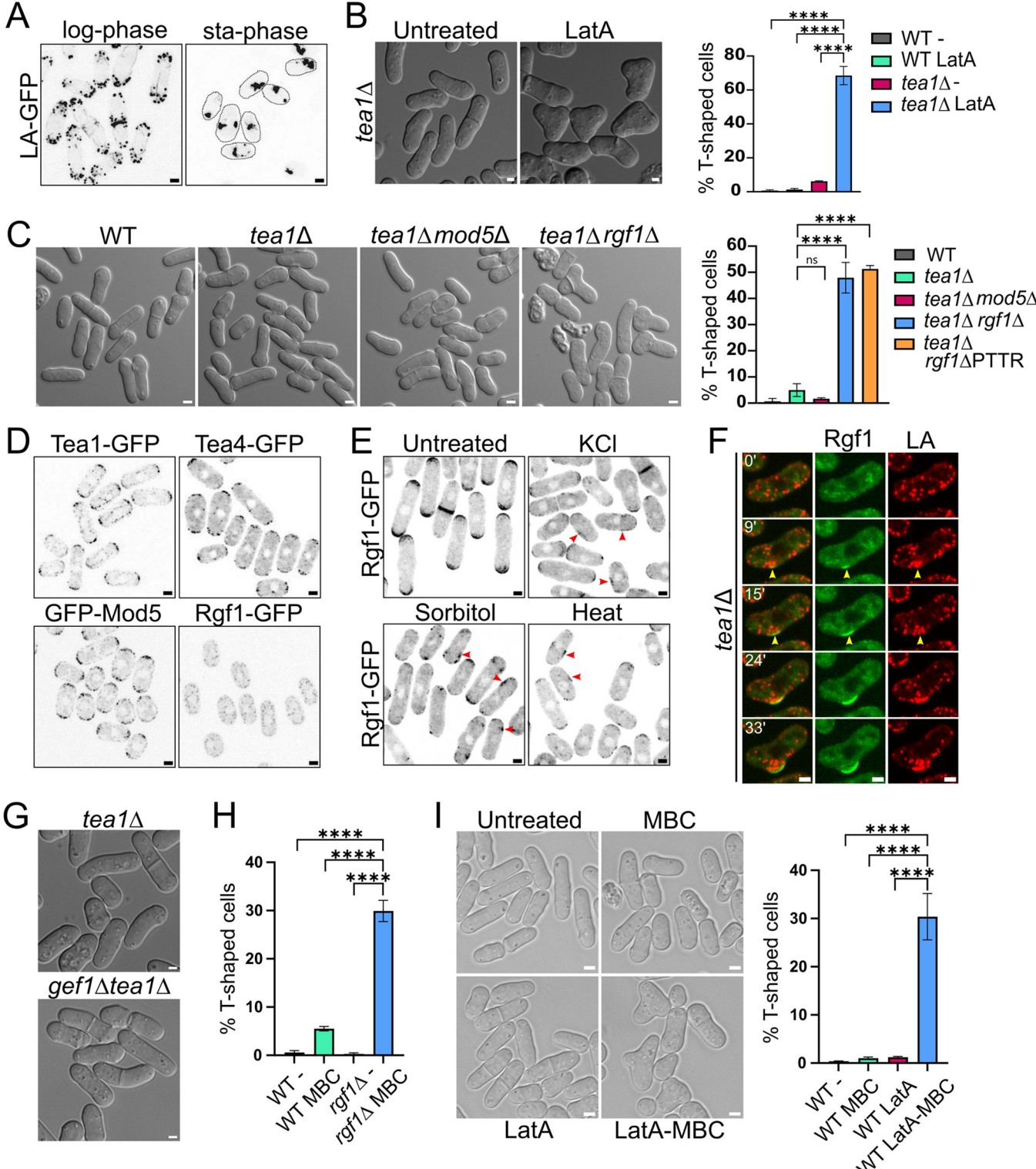

**Fig 5. Rgf1 is part of an actin-dependent machinery that signals growth poles in the absence of the Tea1–Tea4 complex.** (A) Images of LifeAct-GFP (actin) localization in the WT cells growing in liquid medium in the log-phase or after 3 days in the stationary phase (sta-phase). The maximum-intensity projection of 6 Z-slides (0.5 μm step-size) of fluorescence is shown. (B) Morphology and quantitation of the T-shaped WT and *tea1Δ* cells treated for 2 h with DMSO (untreated) or 50 μm of Latrunculin A (LatA) and then washed to allow growth for 3 h. The graph represents the mean ± SD of >200 cells from 3 independent experiments. (C) Morphology and quantitation of the T-shaped cells in the WT, *tea1Δ*, *tea1Δ mod5Δ*, *tea1Δ rgf1Δ*, and *tea1Δ rgf1*ΔPTTR cells grown to log phase in YES liquid medium at 28°C. The graph represents the mean ± SD of >500 cells from 4 independent experiments. (D) Representative images of

Tea1-GFP, Tea4-GFP, GFP-Mod5, and Rgf1-GFP localization in WT cells in the stationary phase after 3 days of growth in liquid medium. The maximum-intensity projection of 6 Z-slides (0.5 μm step-size) of fluorescence is shown. (E) Rgf1-GFP localization in WT cells growing in liquid medium untreated or treated with KCl 0.6 M, sorbitol 1.2 M or 37°C (heat) for 1 h. The maximum-intensity projection of 6 Z-slides (0.5 μm step-size) of fluorescence is shown. The arrowheads point lateral accumulation of Rgf1-GFP. (F) LifeAct-mCherry (actin in red) and Rgf1-GFP localization (green) in *tea1*Δ cells treated with KCl 0.6 M for 1 h and then washed and allowed to grow without stress for the indicated times. The maximum-intensity projection of 4 Z-slides (0.6 μm step-size) of fluorescence is shown. The arrowheads point to lateral accumulation of Rgf1-GFP and actin (LA). (G) Morphology of the *tea1*Δ and *gef1*Δ*tea1*Δ cells growing in YES liquid medium. (H) Quantitation of the T-shaped cells in WT and *rgf1*Δ cells treated with DMSO (-) or with MBC (50 μg/ml) for 4 h. The graph represents the mean ± SD of >200 cells from 3 independent experiments. (I) Morphology and quantitation of the T-shaped cells in WT cells treated with DMSO (WT -), MBC 50 μg/ml (WT MBC) for 4 h, 50 μm of LatA for 2 h, and then washed and allowed to grow with DMSO for 4 h (WT LatA) or first treated with LatA, washed, and then treated with 50 μg/ml of MBC for 4 h (WT LatA-MBC). The graph represents the mean ± SD of >200 cells from 3 independent experiments. Statistical significance was calculated using a two-tailed unpaired Student's *t* test. ****$P < 0.0001$; ns = nonsignificant. Scale bar, 2 μm. The data underlying the graphs shown in the figure can be found in S1 Data. SD, standard deviation; WT, wild type.

patches (Fig 5E). Then, we followed actin reorganization and Rgf1 localization during recovery from osmotic stress in wild-type and *tea1*Δ cells. After relieving the stress, wild-type cells quickly re-concentrated both actin and Rgf1 at the poles (S5E Fig and S4 Movie). However, in *tea1*Δ cells actin and Rgf1 frequently localized at the lateral cortex precisely at points where a branch began to grow (Fig 5F and S5 Movie).

To find out whether other proteins of the growth machinery also define the growth sites or if it is a specific characteristic of Rgf1, we used a mutant lacking *gef1*. Gef1 is a GEF of Cdc42 involved in polarized growth and undergoes a similar relocation from poles to lateral patches under stress, where it is required to activate Cdc42 [52,53]. Interestingly, the *gef1*Δ*tea1*Δ double mutant displayed a similar percentage of T-shaped cells as the *tea1*Δ mutant, indicating that in the absence of Tea1, Gef1 is not necessary for marking the poles, in contrast to Rgf1–Rho1 (Figs 5G and S5F).

Taken together, these results indicated that Rgf1 localization at the cell tips depends on a properly polarize actin cytoskeleton but it is independent on MTs. Thus, both actin concentration and Rho1 activation at the poles are necessary events to define these locations as growth sites.

## Tea1–Tea4-MTs and Rgf1-Rho1-actin define 2 parallel pathways to restrict growth to the cell tips

Because stresses that induce actin disorganization and Rgf1 delocalization at the poles increase the percentage of T-shaped cells in the *tea1*Δ mutant (Figs 5E, S5A and S5B), we asked whether the absence of Rgf1 by itself acts as a stressor, thereby increasing the number of branched cells in the double mutant. In this regard, it has been demonstrated that treatment with LatA causes a delocalization of proteins necessary for polarized growth, such as the active form of Cdc42. The dispersal of Cdc42 from growth sites depends on the activation of MAP kinase Sty1 while it is independent of actin delocalization per se [52,54]. Our group described years ago that while Rgf1 stimulates Pmk1 MAPK pathway activity in response to certain stresses, it was not involved in the activation of Sty1, neither under stress nor in basal conditions [38]. Accordingly, we observed no increase in the nuclear localization of Sty1, nor in the phosphorylation of the transcription factor Atf1 in the *rgf1*Δ mutant (S5G and S5H Fig), as would be expected if Sty1 were active [55]. Unlike delocalization of the active form of Cdc42 (marked by CRIB-GFP) [52,54], Rgf1 delocalization after stress does not depend on Sty1 (S5I Fig). Moreover, the number of cells forming T-shapes in the absence of both Rgf1 and Tea1 does not decrease upon the elimination of Sty1 (S5J Fig). Taken together, these results indicate that the absence of Rgf1 does not activate the Sty1 pathway (stress-activated pathway, SAP), nor is the SAP important for the loss of polarity that occurs in the *rgf1*Δ*tea1*Δ double mutant.

An alternative explanation for the observed increase in branched cells when both Rgf1 and Tea1 were absent is that Tea1 and Rgf1 participate in parallel pathways to define the sites of

cell growth, each one necessary when the other is not present. Thus, we wondered whether chemical ablation of the pathway that delivers Tea1 and Tea4 to the cell tips in a *rgf1Δ* background would yield a similar result. When we treated the *rgf1Δ* mutant with MBC, approximately 30% of the cells exhibited a branched phenotype compared with approximately 5% of the wild-type cells (Fig 5H). Therefore, in the absence of Rgf1 (which lacks the "actin-dependent signal" necessary to recognize the polarity growth zones) and MTs, the imposition of stress is not necessary for branching to occur. The previous results suggest that there could be 2 parallel pathways for positioning the growth poles: one dependent on MTs and the Tea1–Tea4 markers, and another dependent on actin and Rgf1–Rho1. To mimic the elimination of both signaling pathways by chemical treatment, we treated wild-type cells in the log phase with LatA for 2 h (to block actin-dependent signaling), washed them, and exposed them to MBC for 4 h (to remove MTs) to analyze the number of T-shaped cells. Only after the treatment with both, LatA and MBC, approximately 35% of the cells showed branches (Fig 5I). Interestingly, cables and patches similarly contribute to polarized growth in the absence of MTs (S5K Fig).

Our results indicate that Rgf1–Rho1 and Tea1–Tea4 are part of the same complex, with similar functions to delimit the sites of polarized growth of *S. pombe*. We propose that Rgf1 and Rho1 could activate an actin-dependent pathway that instructs cells to grow at the tips regardless of the classical polarity markers.

## Discussion

Cross-talk between MTs and the actin cytoskeleton is crucial for various cellular processes, including asymmetric cell division, the establishment of cell growth zones, and cell migration. In fission yeast, MTs deliver the Tea1–Tea4 complex to the cell tips, where actin concentrates to promote growth. Tea1–Tea4 act as end markers, defining the organization of cell-growth zones and, consequently, the direction of growth. While it is known that these polarity markers are not essential for organizing a growth zone, they become critical for placing the growth zone correctly, especially under stress conditions. However, certain questions remain unanswered, such as how and why the Tea1–Tea4 complex remains linked to the PM once the growth direction is established and how cells mark their tips in the absence of Tea1. In the present study, we addressed some of these questions by demonstrating that Rgf1 functions as a molecular link between the Tea1–Tea4 complex and the PM. Through Rho1 activation, Rgf1 stabilizes Tea4 at the cell ends, promoting its accumulation. Additionally, we described an alternative actin-dependent mechanism, driven by Rgf1 and Rho1, for marking the poles independently to the known MT- and Tea-dependent pathway.

### Rgf1 (Rho1 GEF) activity toward Rho1 is required for stable accumulation of Tea4 at the cell ends

Failure to accumulate Tea4 at the cell cortex in the *rgf1Δ* cells is independent of the movement of Tea4 riding on the tips of polymerizing MTs. Wild-type and *rgf1Δ* cells exhibited similar rates of movement for Tea4-GFP dots from the middle of the cell to the cell ends (Figs 1E and S1G). However, once the Tea4-GFP dots had reached the cell ends, their fluorescence faded in the *rgf1Δ* cells. The refill mechanism responsible for keeping Tea4 stacked at the growing pole depends on correct binding of Rgf1 to the PM and on Rgf1 catalytic activity toward Rho1. We provided evidence that Rgf1 physically binds the polarity marker complex through its interaction with Tea4 (Fig 3B–3D). Moreover, Rgf1 binds to PM phospholipids, likely through its PH domain interacting with membrane PI4P (Fig 3E). The physical association between Rgf1 and PM is crucial for the localization of Tea4 and Tea1 at the cell poles. Indeed, a mutant of Rgf1 lacking the membrane-binding domain, which binds Tea4 in vitro, exhibited similar polarity

defects as the null mutant (Figs 4E and S4G). Additionally, a catalytic mutant (*rgf1-ΔPTTR*) also shows impaired Tea4 anchoring. The Rgf1-ΔPTTR protein binds Tea4 in vitro, localizes to the growing end and is catalytically deficient [38], indicating that Rho1 activity is required to maintain the Tea1-4 complex stable at the pole. However, while Tea4 correct localization is essential, simply targeting Tea4 to the cell poles is not sufficient to rescue growth site selection defects in the absence of Rgf1, especially when MTs are disrupted by MBC treatment (Fig 4F). In this situation, Tea4-GBP-CRIB-GFP may be insufficient to maintain Tea1 localized to the pole as well. In cells treated with MBC, Tea1's absence at the poles could mimic the effects of a double mutant, *rgf1Δ tea1Δ*, because MTs are necessary for the continuous supply of Tea1 to the poles. Without this fresh supply, even if Tea4 is localized correctly, its functional output might be compromised.

Furthermore, the tandem Rgf1–Rho1 also affects Tea1 functions, promoting MT catastrophe once they reach the cell pole or restricting growth sites after refeeding (Figs 1G, 1H and 2B). To maintain polarity markers stably at the poles, Rgf1 functions together with Mod5. The loss of Tea4 localization at the poles is partial in the *rgf1Δ* and *mod5Δ* single mutants and complete when both mutations are combined, causing a defect in polarity even in cells with a continuous supply of Tea1–Tea4 (Fig 2B and 2C). It is likely that Mod5 and Rgf1 anchor the Tea1-4 complex to the PM through different proteins: Mod5 interacts with Tea1 [17], while Rgf1 interacts with Tea4. This mechanism ensures double anchoring of the Tea1–Tea4 complex to the PM; thus, when one of these connections is lost, the other remains available. Only when both are missing do the polarity markers completely lose their connection with the PM, mimicking the behavior of the *tea1Δ* mutant.

## Rgf1–Rho1 are involved in defining the growth sites

Another question that must be addressed is how cells detect their growth sites in the absence of Tea markers. Here, we proposed that Rho1 activation by Rgf1 and actin are behind this process. We observed 2 ways to induce the formation of branched cells in the *tea1Δ* background: one in the presence of certain stresses, "the stress pathway" and the other in the absence of *rgf1*, "Rgf1 depletion" (Figs 5C and S5B). The loss of polarity induced by "the stress pathway" also leads to disorganization of the actin cytoskeleton and, consequently, Rgf1 delocalization. Therefore, both pathways lead to insufficient activation of Rho1 at the poles.

When the *tea1Δ* cells recover from stress, Rgf1 and actin appear simultaneously at sites were ectopic growth occurs (Fig 5F). The interdependence of actin and Rgf1 localization (S5C and S5D Fig) [30] suggests a positive feedback loop between Rho1 activation and actin organization at the growth sites. Therefore, Rgf1–Rho1 would act as Tea markers, defining growth sites without directly promoting growth, given that the *tea1Δrgf1Δ* double mutant retains the ability to grow, albeit at the wrong places. Interestingly, in the absence of Tea1 and Gef1 (Cdc42 GEF), which is also involved in polarized growth and relies on actin for proper localization [52,53,56], cells do not form branches, suggesting that the activation of Rho1 specifically defines the growth sites in the absence of Tea1–Tea4. Recent findings also highlight the redundant role of the glucan synthase Bgs1 and polarity markers in polarized growth [57]. The localization of actin patches and sterol-rich domains at the PM depends on Bgs1 function. However, when both the Tea1–Tea4 complex and Bgs1 were absent simultaneously, there was a complete loss of growth polarity. This resulted in cells taking on a spherical shape rather than the characteristic T-shaped morphology (consequence of loss polarity position) observed when Rgf1 and Tea1–Tea4 were lacking. This suggests that Bgs1 and Rgf1/Rho1 play distinct roles in regulating the polarity process.

The interaction between Tea4 and Rgf1–Rho1 indicates their involvement in the same complex, with each protein essential for the accumulation of the other at the poles. Rgf1 together

with Mod5 acts to link Tea4 to the membrane, and Tea1-4 assists in returning Rgf1 to the poles after stress-induced actin disorganization (Figs 5F and S5E), establishing a functional link between MTs and actin cytoskeletons (Fig 6A).

We showed that Rgf1 and Rho1 are required to maintain an actin-dependent signal that preserves the identity of the cell poles, which becomes evident in the absence of the Tea markers. This would explain why the *rgf1Δtea1Δ* double mutant forms T-shaped cells in the log phase, without stresses or refeeding treatments. We propose that there are 2 different pathways to choose the growth sites under different environmental conditions: the canonical pathway dependent on MT and Tea1–Tea4 and, a novel pathway dependent on actin and Rgf1–Rho1. Various combinations of mutants and/or chemical ablation of one component from each pathway (Tea1–Tea4–MT and Rgf1–Rho1–actin) at once, lead to comparable outcomes. For example, chemical actin depolymerization in the *tea1Δ* cells induces branching (Fig 5B), whereas treatment of the *rgf1Δ* cells with MBC to prevent the constant supply of Tea markers generates T-shaped cells as well (Fig 5H). Furthermore, wild-type cells treated to transiently depolarize actin and subsequently prevented for refeeding of polarity markers to the tips (via MT de-polymerization) also experience difficulties in detecting the cell poles (Fig 5I). Thus, cells would have different pathways to maintain their cylindrical morphology even if one of them is challenged by internal or external conditions. When the continuous supply of Tea1 from MT is disrupted (Fig 6B), cells could use an additional cue provided for a different cytoskeletal polymer, actin. Similarly, when actin becomes disorganized, for example, in the transition from monopolar to bipolar growth (NETO) or in *rgf1Δ* mutant (defective in actin polarization), cells would count on polarity markers transported by MTs (Fig 6C). Only when both, the actin and MT cytoskeletons are compromised, fission yeast cells lose their cylindrical shape (Fig 6D). This sophisticated regulation highlights the importance of maintaining cell morphology throughout the cell cycle and under changing environmental conditions.

## Materials and methods

### Media, reagents, and genetics

*S. pombe* strains were streaked on plates of complete yeast growth medium (YES) or selective medium (EMM) supplemented with the appropriate requirements [58], and incubated at 28˚C until colonies formed. For each biological replicate, a single colony was used to inoculate 5 ml of the respective liquid media. Cultures were incubated at 28˚C overnight with shaking (200 rpm). Each overnight culture was subsequently used as a seed culture to inoculate fresh media. Fresh cultures were next grown at 28˚C, 200 rpm, to OD 600 = 0.5–0.6 at the time of harvest. Crosses were performed by mixing the appropriate strains directly on sporulation medium plates. Recombinant strains were obtained by tetrad analysis or the "random spore" method. For overexpression experiments using the nmt1 promoter, cells were grown in EMM containing 15 μm thiamine up to the logarithmic phase. Then, the cells were harvested, washed 3 times with water, and inoculated in fresh medium (without thiamine) at an optical density at 600 nm ($OD_{600}$) of 0.01 for 18 to 20 h. Two-hybrid interaction was tested with YNB medium lacking histidine in *Saccharomyces cerevisiae* strain AH109 (Clontech). For refeeding experiments, we have incubated the cells at 28˚C for 3 days in liquid EMM, to reach stationary phase, and diluted them 1:20 in fresh medium to let them grow for 3 h at 28˚C.

### Plasmid and DNA manipulations

Plasmids used in this study are listed in S2 Table. pREP4x-HArho1 (with thiamine-repressible *nmt1* promoter) and pGEX-C21RBD plasmids (rhotekin-binding domain) kindly provided by Pilar Pérez (Instituto de Biología Funcional y Genómica, Salamanca, Spain) were used to

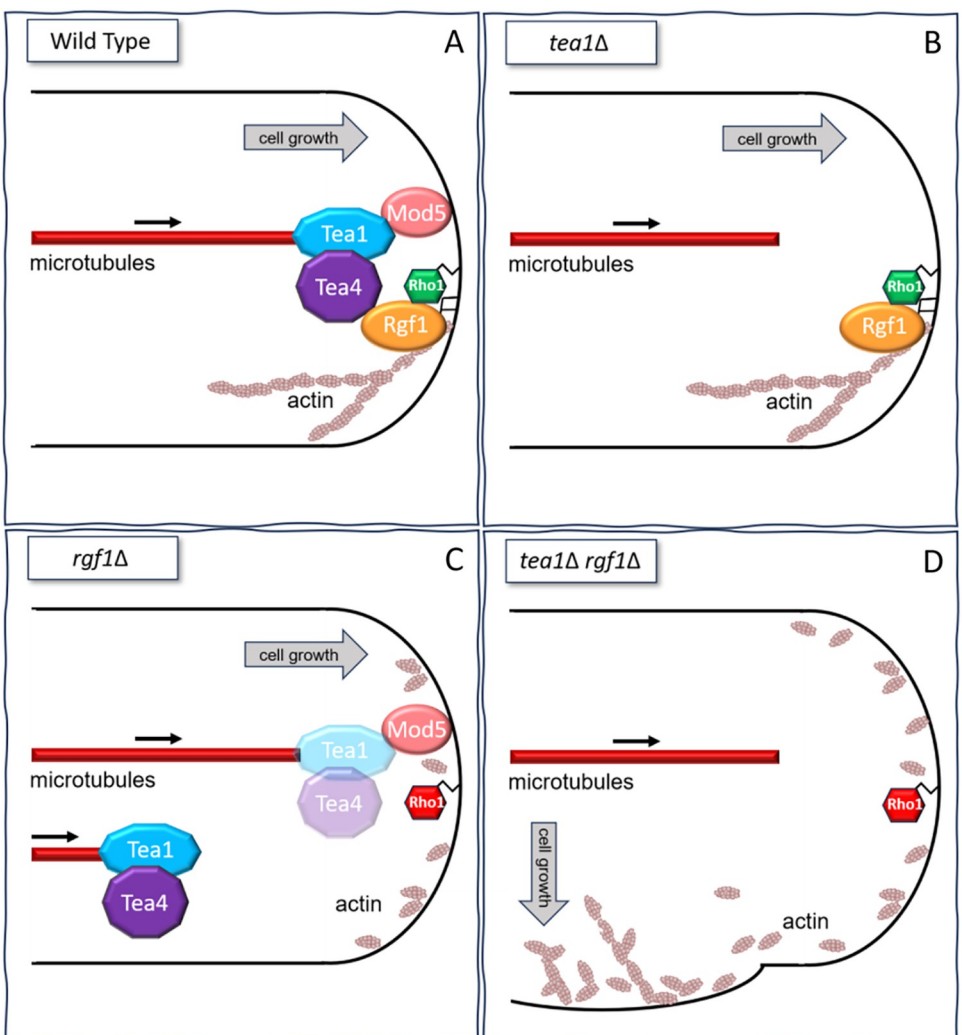

**Fig 6. Rgf1-Rho1 functions as a molecular link between Tea4 and the PM and marks the growth sites in an actin-dependent manner.** (A) In wild-type cells MT deliver Tea1 and Tea4 to the cell poles, where they bind to the membrane through their interaction with Mod5 and Rgf1, respectively. Rgf1, in turn, interacts with the PM due to its affinity for the phospholipid PI4P and activates Rho1, promoting proper actin cytoskeleton polarization at the cellular tips. (B) Cells lacking Tea1 still recognize the cell poles correctly because Rgf1 is localized in these regions, where it activates Rho1, thus allowing the maintenance of polarized actin. (C) In *rgf1*Δ cells Tea1–Tea4 partially disappears from the pole, although a remnant remains attached to Mod5. Rho1 would be inactive, leading to actin cytoskeleton disorganization. However, the continuous supply of Tea1–Tea4 from MTs persists, allowing the cells to grow correctly. (D) When cells lack *tea1* and *rgf1*, they lose both pathways that allowed them to distinguish the tips. First, there is no continuous supply of polarity markers towards the cell ends to mark the growth sites. Second, Rgf1 is not at the pole to activate Rho1, leading to actin disorganization. The elimination of both pathways causes the cells to direct their growth towards incorrect locations, where growth factors probably accumulate. MT, microtubule; PM, plasma membrane.

detect Rho1-GTP levels. To express proteins in *Escherichia coli*, we used a pGEX-2T plasmid that contains a GST to tagged genes at the 5′ end. pGEX-*rgf1* (pGR152) and pGEX-*tea4* (pGR129) were made by inserting the entire ORF of *rgf1* (without introns) or *tea4* in frame into pGEX-2T and purified to be used in pull-down assays. To perform lipid strips assays, we constructed the plasmids pGR128 that contains the ORF of *rgf1* (from amino acid 1–922) without the last 412 amino acids containing the CNH domain and pGR138 that contains the ORF of *rgf1* (from amino acid 1–722) without the last 612 amino acids containing the PH and CNH

domains. For two-hybrid experiments, we constructed the plasmids pGAD-*tea1* (pGR135), pGAD-*tea4* (pGR106), and pGBK-*rgf1* (pRZ97) where the entire ORF of the corresponding gene was inserted in frame into the pGADT7 (GAL4 activation domain) or pGBKT7 (GAL4 binding domain) plasmid (Clontech). For localization of the Rgf1 lacking the PH domain (pGR145), pGR41 (pJK148-*rgf1*-GFP) was mutagenized, followed by the elimination of the entire PH domain between residues 772–1.003. To construct the pact-*rgf1*ΔPH-GFP strain, the promoter of *rgf1* from the plasmid pGR145 (pJK148-*rgf1*ΔPH-GFP) was replaced with 990 nucleotides from the promoter of the *act1* gene of *S. pombe*. To construct the *rgf1*ΔPHΔPTTR strain, plasmid pGR156 (pJK148-*rgf1*ΔPH) was mutagenized to eliminate the 4 amino acids from the GEF catalytic site.

## Protein extracts and immunoblot analysis

*S. pombe* cultures (5 ml) at an $OD_{600}$ of 0.5 were pelleted just after the addition of 10% TCA and washed in 20% TCA. The pellets were resuspended in 100 µl 12.5% TCA with the addition of glass beads and lysed by vortexing for 5 min. Cell lysates were pelleted, washed in iced acetone, and dried at 55˚C for 15 min. Pellets were resuspended in 50 µl of a solution containing 1% SDS, 100 mM Tris–HCl (pH 8.0), and 1 mM EDTA. Samples were electrophoretically separated by SDS-PAGE (4%–15% MiniProtein Gel, BioRad) and immunodetected with anti-GFP (Living Colors, RRID:AB10013427) and anti-mouse (Bio-Rad, AB_11125547) antibodies. As a loading control, we used monoclonal antitubulin antibodies (Sigma, RRID:AB_477579). For Atf1 detection, we used an anti-Atf1 antibody kindly provided by Elena Hidalgo (Universitat Pompeu Fabra, Barcelona, Spain).

## Immunoprecipitations and pull-down assays

Immunoprecipitation assays were performed as described previously with some modifications [59]. Briefly, logarithmic cultures were pelleted and re-suspended in lysis buffer (10 mM Tris–HCl (pH 7.5), 150 mM NaCl, 0.5 mM EDTA, 0.5% NP40, containing 100 µm PMSF, leupeptin, and aprotinin) and lysed in a cryogenic grinder. Lysates were centrifuged for 5 min at 6,000 g, and then 10 µl of GFP-Trap magnetic beads (Chromotek) was added to the supernatants and incubated for 1 h at 4˚C. Immunoprecipitates were washed 3 times with dilution buffer (10 mM Tris–HCl (pH 7.5), 150 mM NaCl, and 0.5 mM EDTA). Proteins were released from immunocomplexes by boiling for 5 min in sodium dodecyl sulfate (SDS) loading buffer. Samples were separated by SDS–polyacrylamide gel electrophoresis (4%–15% Mini-Protean TGX gels, Bio-Rad) and detected by immunoblotting with polyclonal anti-HA (Roche, RRID:AB_514506) or anti-GFP antiserum (Living Colors, RRID:AB_10013427). For pull-down assays, first GST-tagged Rgf1, Tea4, or C21RBD (rhotekin-binding domain to detect Rho1-GTP levels) was purified from *E. coli* (BL21). The fusion proteins were produced by adding 0.5 mM IPTG at 18˚C overnight (or 3 h at 28˚C for C21RBD). Cells were sonicated and proteins immobilized on glutathione-Sepharose (GS) 4B beads (GE-Healthcare). After incubation for 1 h, the beads were washed several times, and the bound proteins were analyzed by SDS-PAGE and stained with Coomassie brilliant blue. Pull-down assays were performed as described previously [60]. In brief, extracts from the indicated strains were obtained by using 500 µl of lysis buffer (50 mM Tris–HCl (pH 7.5), 20 mM NaCl, 0.5% NP-40, 10% glycerol, 0.1 mM dithiothreitol, and 2 mM $Cl_2Mg$, containing 100 µm PMSF, leupeptin, and aprotinin) and lysed in a cryogenic grinder. Cell extracts (2 to 3 mg of total protein) were incubated with 2 to 10 µg of GST-tagged protein coupled to GS beads for 2 h, washed 4 times with lysis buffer, and blotted with an anti-HA or anti-GFP antibody. Protein levels in whole-cell extracts (80 µg of total protein) were monitored by western blot. Tubulin and GST were used as loading controls.

## Lipid strip overlay assays

Lipid strip overlay assays were performed as described previously [61] using lipid strip membranes (p-6002, Echelon). Strips were blocked with 3% fatty acid-free bovine serum albumin (BSA; Sigma) in TBS-T (10 mM Tris–HCl (pH 8.0), 150 mM NaCl, and 0.1% Tween-20; TBST-BSA) at room temperature for 1 h and then incubated for 2 h with 1 μg/ml GST, GST-Rgf1, or GST-Rgf1ΔPH in TBST-BSA. The strips were then washed 3 times with 5 ml of TBST-BSA and incubated with anti-GST horseradish peroxidase-conjugated antibody (GE Healthcare, RRID:AB_771429) diluted in TBST-BSA. Bound protein was detected using an enhanced chemoluminescence detection kit (BioRad). GST, GST-Rgf1, and GST-Rgf1ΔPH were expressed in *E. coli* (BL21) and purified with GS beads (GE-Healthcare) according to the manufacturer's instructions as described above. Once attached to the beads, they were washed and eluted with 200 μl of elution buffer (100 mM Tris–HCl (pH 8.0), 120 mM NaCl) with 20 mM of L-glutathione reduced (Sigma) freshly added for 30 min at 4˚C. Aliquots were frozen in liquid nitrogen with 15% of glycerol and stored at −80˚C until use.

## Microscopy

Wet preparations were observed with an Andor Dragonfly 200 Spinning-disk confocal microscope equipped with a sCMOS Sona 4.2B-11 camera (Andor) and controlled with the Fusion 2.2 acquisition software, or a Personal Deltavision (Applied Precision, LLC) microscope equipped with a CoolSNAP HQ2 camera (Photometrics) and controlled with softWoRx Resolve 3D. Depending on the experiment, a single focal plane at the center of the cell or a stack of 4 to 6 images covering the entire volume of the cell (Z-series) with a spacing of 0.5 to 0.6 μm were captured, and the maximum projection was generated.

To analyze protein dynamics, time-lapse experiments were performed with cells in μ-Slide 8-well (Ibidi) coated with soybean lectin (Sigma Aldrich) and imaged at the indicated times using an Andor Dragonfly Spinning-disk confocal microscope. We used the ImageJ 1.53t software to calculate the relative fluorescence intensity of each protein at the cell tips. We have designed an ImageJ macro to automatically select the fluorescence regions (ROI manager) of the cell poles and to measure the fluorescence intensity of 40 to 130 cells. We used the integrated density value of each tip of the different strains versus the average of the integrated density of the wild-type strain to calculate the relative fluorescence levels. To create kymographs, we drew a line from the cell's center to the pole along a MT on a time-lapse movie and utilized the KymographBuilder plugin of the ImageJ software. For SRRF images of Tea4-GFP (green) and mCherry-Atb2 (red) in "head-on" cell tips, cells were mounted onto a μ-Slide pre-coated with lectin. Bound cells found to be frontally arrayed ("head-on") were visually selected for imaging using the SRRF module of the Andor Dragonfly Confocal microscope. Fifty repetitions of one focal plane were taken for each time point. The time projection of the 3 images at different time points is shown to follow Tea4 cluster movement.

Calcofluor white (Blankophor BBH, Bayer Corporation) staining was performed by adding 1 μl of a stock solution (2.5 mg/ml) to 500 μl of samples for 20 s, followed by a wash with phosphate-buffered saline (PBS).

## Quantification and statistical analysis

Statistical analysis and graphs were generated using GraphPad Prism Software version 9.5.1. To compare 2 conditions, a two-tailed unpaired Student's *t* test was applied to determine statistical significance (as detailed in the figure legends). To compare multiple conditions, a two-way ANOVA test was applied to determine statistical significance (as detailed in the figure legends). $P < 0.05$ was considered significant. The graphs show the mean ± standard deviation

(SD) of the indicated data. Asterisks represent the following: $^*P < 0.05$, $^{**}P < 0.01$, $^{***}P < 0.001$, $^{****}P < 0.0001$.

## Supporting information

**S1 Table. List of yeast strains used in this study.**
(DOCX)

**S2 Table. List of plasmids used in this study.**
(DOCX)

**S1 Fig. Rgf1 is required for proper localization of the Tea1-Tea4 complex at the cell tip.**
(A) The graphic represents the mean ± SD of the relative fluorescence intensity of Tea4-GFP measured at the growing and non-growing ends of WT cells ($n > 150$ for each end) and *cdc10-129* cells ($n > 150$ for each end) grown for 4 h at 37˚C. Calcofluor staining was used to differentiate the growing poles from the non-growing poles. (B) Wild-type cells expressing rgf1-GFP were stained with Calcofluor white (20 μg/ml) to spot areas of growth. The arrows indicate the localization of Rgf1 at the growing tip in monopolar cells. (C) Maximum projection images of the cells from Fig 1C show the merged localization of Tea4-GFP (green) and Tea1-tomato (red) in wild-type (WT) and *rgf1Δ* cells (left panels). The fluorescence intensity profiles for green and red channels along a line across the poles, covering the cell width, are shown for cells 1 and 2 of WT and *rgf1Δ* cells, respectively, in the right panels. Scale bar, 2 μm. (D) GFP fluorescence in cells expressing *tea1-GFP rgf1*$^+$ and *tea1-GFP rgf1Δ* grown to log-phase at 28˚C in YES medium. The maximum-intensity projection of 6 Z-slides (0.5 μm step-size) of Tea1-GFP fluorescence is shown. The graphic represents the mean ± SD of the relative fluorescence intensity of Tea1-GFP measured at the growing and non-growing ends of WT ($n > 80$ for each end) and *rgf1Δ* ($n > 80$ for each end) cells. (E) Quantitation of the number of Tea4 dots associated to the MTs in WT and *rgf1Δ* strains per cell. The mean ± SD of $>80$ cells is shown. (F) The WT and *rgf1Δ* cells expressing *tea4*-GFP were treated with the translation inhibitor cycloheximide (CHX, 100 μg/ml) for the indicated times. Proteins were visualized by western blot with antibodies against GFP (Tea4) or tubuline (Tub), as a loading control (upper). The graphic represents the quantification of Tea4 levels at different times (hours) after the treatment relative to time 0, which was assigned a value of 1 (bottom). (G) The graphics show the MT polymerization (left) and depolymerization (right) rates in WT and *rgf1Δ* cells. The mean ± SD of $>75$ cells is shown. Statistical significance was calculated using two-tailed unpaired Student's *t* test. $^*P < 0.05$; $^{****}P < 0.0001$; ns = nonsignificant. The data underlying the graphs shown in the figure can be found in S1 Data.
(PDF)

**S2 Fig. Rgf1 cooperates with Mod5 in Tea4 anchoring to the cellular poles.** (A) Representative images of the indicated strains producing Tea1-GFP. The maximum-intensity projection of 6 Z-slides (0.5 μm step-size) is shown. Scale bar, 2 μm. The graph represents the mean ± SD of the relative fluorescence intensity of Tea1-GFP measured at the cell tips in the WT, *rgf1Δ*, *mod5Δ*, and *rgf1Δ mod5Δ* cells ($n > 100$). WT levels were used for normalization. (B) Cell morphology of the indicated strains after refeeding treatment (-MBC). (C) Cell morphology of the indicated strains after 4 h at 36˚C in YES liquid medium. Scale bar, 2 μm. $^{****}P < 0.0001$. The data underlying the graphs shown in the figure can be found in S1 Data.
(PDF)

**S3 Fig. Rgf1 interacts with the cell end marker Tea4 and binds to phosphatidylinositol-4-phosphate through its PH domain.** (A) Colocalization of Rgf1 and Tea4. Super-resolution

radial fluctuations (SRRF) images of WT cells producing Tea4-GFP endogenously (green) and Rgf1-tdTomato from a plasmid under the control of its own promoter (red). (B) Coprecipitation of Rgf1 and Tea1. Cell extracts from cells producing Tea1-GFP, Rgf1-HA, and Tea1-GFP and Rgf1-HA were precipitated with GFP-trap beads and blotted with anti-HA or anti-GFP antibodies (co-immunoprecipitation and immunoprecipitation). Western blot was performed on total extracts to visualize total Tea1-GFP and Rgf1-HA levels (whole cell extracts). The data underlying the graphs shown in the figure can be found in S1 Data.
(PDF)

**S4 Fig. Tea4 accumulation at the cell ends depends on Rgf1 anchoring to the PM and Rho1 activation.** (A) The graphic represents the mean ± SD of the relative fluorescence intensity of Tea4-GFP measured at the growing and non-growing ends of the strains indicated ($n > 100$ for each end). Calcofluor staining was used to differentiate the growing poles from the non-growing poles. (B) Protein extracts from cells producing $rgf1^+$-GFP, $rgf1\Delta$PH-GFP, and pact-$rgf1\Delta$PH-GFP were analyzed by western blot with an anti-GFP antibody to visualize Rgf1 levels. An anti-tubulin antibody was used as a loading control. (C) Extracts from cells producing Rho1-HA (pREP4X-Rho1-HA) in the WT, $rgf1\Delta$, $rgf1\Delta$PH, and pact-$rgf1\Delta$PH-GFP cells were pulled down with GST-C21RBD and blotted against anti-HA antibody (Rho1-GTP). Total Rho1-HA was visualized by western blot (WCE). The relative units indicate the fold-differences in Rho1 levels in the mutants compared with the WT strain, with an assigned value of 1 (bottom) from 2 independent experiments. (D) Maximum-intensity projection of 6 Z-slides (0.5 μm step-size) of representative fluorescence images of WT, $rgf1\Delta$PH-GFP, and pact-$rgf1\Delta$PH-GFP cells. (E) The graphic represents the mean ± SD of the relative fluorescence intensity of Tea4-GFP ($n > 120$) measured at the cellular tips of the WT, $rgf1\Delta$PH, and pact-$rgf1\Delta$PH. Statistical significance was calculated using a two-tailed unpaired Student's $t$ test. (F) Representative images of Rgf1-GFP and Rgf1ΔPTTR-GFP localization. Cells were stained with calcofluor white (20 μg/ml) to spot areas of growth. Scale bar, 2 μm. (G) The percentage of cells WT, $tea1\Delta$, $rgf1\Delta$, $rgf1\Delta$PH, pact-$rgf1\Delta$PH, and $rgf1\Delta$PH-ΔPTTR cells forming branches 3 h after release to growth after 3 days in stationary phase, in the absence and in the presence of MBC (50 μg/ml). The mean ± SD of >200 cells from 3 independent experiments is shown. Statistical significance of each strain compared to the WT was calculated using a two-way ANOVA test. ****$P < 0.0001$, *$P < 0.05$, ns = nonsignificant. The data underlying the graphs shown in the figure can be found in S1 Data.
(PDF)

**S5 Fig. Rgf1 is part of actin-dependent machinery that signals growth poles in the absence of Tea1–Tea4 complex.** (A) Representative images of LifeAct-GFP (actin) localization in WT cells untreated or treated with KCl 0.6 M, sorbitol 1.2 M, or 37°C (heat) for 1 h. The maximum-intensity projection of 6 Z-slides (0.5 μm step-size) of fluorescence is shown. (B) Quantitation of the T-shaped cells in the $tea1\Delta$ mutant treated with DMSO (Unt.), KCl 0.6 M, sorbitol 1.2 M, or 37°C (heat) for 1 h, then washed and allowed to grow without the drug for 3 h. The graph represents the mean ± SD of >200 cells from 2 independent experiments. Statistical significance was calculated using a two-tailed unpaired Student's $t$ test. (C) Images of WT and $rgf1\Delta$ cells expressing LifeAct-GFP to visualize actin. Maximum-intensity projection of cells ($n = 22$) of the same size from each strain is included to show the distribution of actin patches along the cells. (D) Cells expressing rgf1-GFP and crn1-GFP (actin patches) or mCherry-atb2 (MTs) cultured separately, mixed, and then treated with DMSO, LatA 100 μm, or MBC 50 μm for 15 min. (E) LifeAct-mCherry (actin) and Rgf1-GFP localization in WT cells treated with KCl 0.6 M for 1 h and then washed and allowed to grow without stress for the indicated times. The localization of Rgf1-GFP in $tea4\Delta$ cells under the same conditions is

shown in the lower panel. The maximum-intensity projection of 4 Z-slides (0.6 μm step-size) of fluorescence is shown. (F) Quantitation of the T-shaped cells in the WT, *tea1Δ*, *gef1Δ*, and *tea1Δ gef1Δ* cells grown to log phase in YES liquid medium at 28°C. The graph represents the mean ± SD of >500 cells from 2 independent experiments. Statistical significance was calculated using a two-tailed unpaired Student's *t* test. (G) GFP fluorescence in WT and *rgf1Δ* log-phase cells expressing *sty1-GFP*. Cells either were untreated (upper panels) or treated with 1 mM $H_2O_2$ for 15 min (lower panels). The maximum-intensity projection of 6 Z-slides (0.5 μm step-size) of Sty1-GFP fluorescence is shown. (H) Extracts from wild type (WT-unt) and *rgf1Δ* cells (*rgf1Δ*-unt) were compared with extracts from WT cells treated with 1 mM $H_2O_2$ for 15 min. Proteins were visualized by western blot with antibodies against Atf1. (I) GFP fluorescence in WT and *sty1Δ* log-phase cells expressing *Rgf1-GFP*. Cells were either untreated (upper panels) or treated with 0.6 M KCl for 1 h (lower panels). The maximum-intensity projection of 6 Z-slides (0.5 μm step-size) of Rgf1-GFP fluorescence is shown. (J) The percentage of T-shaped cells (*n* >200) in the indicated strains grown to log phase in YES liquid medium at 28°C. Statistical significance was calculated using a two-tailed unpaired Student's *t* test. ns = nonsignificant. The experiment was repeated 4 times. (K) Quantitation of the T-shaped cells in WT cells treated with DMSO (WT DMSO), MBC 50 μg/ml (WT MBC) for 4 h, 100 μm of CK666 for 2 h to eliminate actin patches and then washed and allowed to grow with DMSO (WT CK) or 50 μg/ml of MBC for 4 h (WT CK-MBC); *for3Δ* cells, which lack actin cables, were treated with DMSO (*for3Δ* DMSO) or MBC 50 μg/ml (*for3Δ* MBC) for 4 h. WT cells treated with 50 μm of LatA for 2 h were washed and then treated with 50 μg/ml of MBC for 4 h (WT LatA-MBC). The graph represents the mean ± SD of >200 cells from 3 independent experiments. Statistical significance was calculated using a two-way ANOVA test. ****$P < 0.0001$; **$P < 0.01$; ns = nonsignificant. Scale bar, 2 μm. The data underlying the graphs shown in the figure can be found in S1 Data.
(PDF)

**S1 Movie. Tea4 and MT dynamics in wild-type cells.** Tea4-GFP (green) and mCherry-Atb2 (red) localization in WT cells. Protein dynamics was followed for 8 min, with pictures taken every 20 s. The maximum-intensity projection of 6 Z-slides (0.5 μm step-size) is shown.
(AVI)

**S2 Movie. Tea4 and MT dynamics in *rgf1Δ* cells.** Tea4-GFP (green) and mCherry-Atb2 (red) localization in *rgf1Δ* cells. Protein dynamics was followed for 8 min, with pictures taken every 20 s. The maximum-intensity projection of 6 Z-slides (0.5 μm step-size) is shown.
(AVI)

**S3 Movie. LifeAct-GFP dynamics and morphology in *rgf1Δ tea1Δ* cells.** LifeAct-GFP (green) and DIC (gray) were superposed to follow actin localization and morphology changes during 120 min (each picture every 2.5 min) in *rgf1Δ tea1Δ* cells growth in YES medium at 30°C.
(AVI)

**S4 Movie. Rgf1 and actin dynamics in WT cells during osmotic stress recovery.** Rgf1-GFP (green) and LifeAct-mCherry (actin in red) localization in WT cells treated with KCl 0.6 M for 1 h and then washed and allowed to grow without stress. Protein dynamics was followed for 42 min, with pictures taken every 3 min. The maximum-intensity projection of 4 Z-slides (0.6 μm step-size) is shown.
(AVI)

**S5 Movie. Rgf1 and actin dynamics in *tea1Δ* cells during osmotic stress recovery.** Rgf1-GFP (green) and LifeAct-mCherry (actin in red) localization in *tea1Δ* cells treated with KCl 0.6 M for 1 h and then washed and allowed to grow without stress. Protein dynamics was followed for 66 min, with pictures taken every 3 min. The maximum-intensity projection of 4 Z-slides (0.6 μm step-size) is shown.
(AVI)

**S1 Raw Images. Uncropped gel images used in this study.**
(PDF)

**S1 Data. Excel spreadsheet containing, in separate sheets for each figure panel, the individual numerical data used in this study.**
(XLSX)

## Acknowledgments

We thank J.C. Ribas, Phong T. Tran, Sophie Martin, Sergio Rincón, Ken Sawin, Paul Nurse, James Moseley, Kathleen L. Gould, César Roncero, Henar Valdivieso, Sergio Moreno, Elena Hidalgo, and Pilar Pérez for sharing strains and plasmids. We also wish to thank Javier Encinar del Dedo and Sergio Rincón for their very helpful comments on the manuscript. We are grateful to Jesús Pinto (IBFG bioinformatics facility) for ImageJ macro used for fluorescence quantification.

## Author Contributions

**Conceptualization:** Patricia Garcia, Yolanda Sanchez.

**Funding acquisition:** Yolanda Sanchez.

**Investigation:** Patricia Garcia, Ruben Celador, Tomas Edreira.

**Methodology:** Patricia Garcia.

**Project administration:** Yolanda Sanchez.

**Supervision:** Patricia Garcia, Yolanda Sanchez.

**Writing – original draft:** Patricia Garcia.

**Writing – review & editing:** Patricia Garcia, Yolanda Sanchez.

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
