## [Editor Report · Decision Letter 0]

8 Jan 2024

Dear Dr Sanchez, 

Thank you for submitting your manuscript entitled "Rgf1 GEF activity toward Rho1 defines a new actin-dependent signal to determine growth sites independently of microtubules and Tea1" for consideration as a Research Article by PLOS Biology.

Your manuscript has now been evaluated by the PLOS Biology editorial staff as well as by an academic editor with relevant expertise and I am writing to let you know that we would like to send your submission out for external peer review.

Once your full submission is complete, your paper will undergo a series of checks in preparation for peer review. After your manuscript has passed the checks it will be sent out for review. To provide the metadata for your submission, please Login to Editorial Manager (https://www.editorialmanager.com/pbiology) within two working days, i.e. by Jan 10 2024 11:59PM.

Kind regards,

Ines

--

Ines Alvarez-Garcia, PhD

Senior Editor

PLOS Biology

---

## [Decision Letter · Decision Letter 1]

19 Mar 2024

Dear Dr Sanchez,

Thank you for your patience while your manuscript entitled "Rgf1 GEF activity toward Rho1 defines a new actin-dependent signal to determine growth sites independently of microtubules and Tea1" was peer-reviewed at PLOS Biology. Please also accept my apologies for the time it has taken us to provide you with a decision. The manuscript has now been evaluated by the PLOS Biology editors, an Academic Editor with relevant expertise, and by two independent reviewers. In addition, the Academic Editor has added in depth comments on the manuscript. In light of the reviews, we would like to invite you to revise the work to thoroughly address the reviewers' reports. 

The reviews are attached below. As you will see, the two reviewers find the manuscript interesting, but they raise several concerns, along with the Academic Editor, that would need to be addressed before we can consider the manuscript for publication. They think that you should investigate Tea1 localisation in addition to Tea4, and overexpress rgf1∆PH to compensate for instability – the reviewers note that it is very difficult to interpret these data. In addition, you should consider the possibility that rgf1∆ elicits stress signaling, and how Rgf1 may be connected to the actin cytoskeleton should be tested a bit more in depth if the stress signaling hypothesis is incorrect. The Academic Editor also thinks that it is not necessary to repeat the Tea1-Rgf1 interaction experiments (Reviewer 1, part of point 3) nor to extend the analysis to the localisation of other proteins (Reviewer 1, point 5). All the other points raised by Reviewer 2 should be addressed with the exception of Point 8.

Given the extent of revision needed, we cannot make a decision about publication until we have seen the revised manuscript and your response to the reviewers' comments. Your revised manuscript is likely to be sent for further evaluation by all or a subset of the reviewers.

**IMPORTANT - SUBMITTING YOUR REVISION**

3. Resubmission Checklist

a) *PLOS Data Policy*

b) *Published Peer Review*

Sincerely,

Ines

--

Ines Alvarez-Garcia, PhD

Senior Editor

PLOS Biology

Reviewers' comments

Rev. 1:

In this manuscript, García et al. analyze the biological relevance of the Rho1 GEF Rgf1 in the establishment of polar growth in fission yeast. The work presents two main conclusions: 1) Rgf1 contributes to Tea4 and Tea1 polarity landmarks anchoring at the cell poles; 2) Two pathways define the growth poles in fission yeast: one dependent on microtubules and Tea1-Tea4 and another dependent on actin and Rgf1-Rho1.

Evidence in multiple organisms supports that precise cell polarization relies on interactions between the microtubule and actin cytoskeletons. In fission yeast, the precise location of polar growth activation is established by depositing the Tea1-Tea4 complex at the cell poles. Without this complex, cells cannot accurately position growth at the cell poles, leading to irregularly shaped cells. Additionally, when growth is reinitiated de novo, it might occur at a site other than the cell tips, resulting in the formation of T-shaped cells (as shown in seminal papers from Nurse, Sawin and Chang laboratories). Polarity establishment depends on Tea1 anchored at the poles by Mod5 and a small amount deposited at the cell ends in association with microtubules ends (Snaith and Sawin, 2003). Simultaneous disruption of both microtubules and actin cables still permits polarized cell growth, although the cells do not correctly position the growth poles (Bendezú and Martin, 2010). Indeed, de novo polarization after starvation depends on the polarized recruitment of sterol-rich domains, which relies on Tea1 and F-actin (Makushok et al., 2016).

The authors of this manuscript build upon these previous observations, some of which were not referenced, to demonstrate that Rgf1-Rho1 play a role in establishing polarity under specific circumstances. The data that leads to and extends point 1 are compelling but not entirely novel. In 2017, Dogdson et al. reported that deleting Rgf1 resulted in a consistent reduction in the fluorescence enrichment of polarity landmarks Tea1, Tea4 and Pom1 at the cell tips (http://dx.doi.org/10.1101/116749). However, this article has not been cited by the authors and is not mentioned in the introduction, despite its relevance to the manuscript's topic. The analysis that leads to point 2 is not that persuasive. It is important to note that while understanding the control of cell polarity is of great interest in the field of morphogenesis, it remains unclear whether the conceptual advances presented in this work are sufficient, given the uncertainty regarding Rgf1's target/s in the control of actin cytoskeleton, and the already previous published data that acknowledge the contribution of both microtubule-dependent signals and actin-dependent signals for the establishment of polarity. In summary, given the lack of mechanistic insight in the latter sections of the paper and the modest advancement in understanding presented in the former, coupled with the numerous reservations regarding data presentation, experimental design, and quantification, I am not able to express great enthusiasm for the paper in its current state. Please refer to the comments below for further specifics regarding the manuscript.

1. Rgf1 is required to locate the Tea1-Tea4 complex properly and to integrate Tea4 in big clusters at the cell tip (Figure 1). As mentioned earlier, reduced localization of Tea1 and Tea4 in rgf1-null cells has already been described. The data are not novel, and previous work has not been appropriately cited.

Figure 1A, B. Authors quantify fluorescence intensity of Tea4-GFP at the cell tips in WT vs rgf1- cells and claim that in rgf1- cells, the signal was visibly diminished. Is the signal reduced, or is it just that in most wild-type cells, Tea4-GFP is localized at the two poles and in rgf1- just in one pole? In panel B, the fluorescence intensity of Tea4-GFP in wild-type cells appears to be similar to that observed in the non-growing pole in rgf1- cells. Can the authors include measurements of wild-type cells to compare fluorescence intensity with the non-growing pole of rgf1- cells? Additionally, it would be interesting to know if other monopolar mutants, distinct from polarisome components, also show reduced Tea4 localization at the growing pole.

Figure 1C. The authors claim that there is a wide co-localization between Tea1 and Tea4 in the absence of Rgf1 and conclude that both proteins displayed similar localization defects. Nevertheless, many red spots corresponding to Tea1-tdT in the merged image can be observed, suggesting that the effect on Tea1 localization is not severe. It is difficult to conclude from the images provided that Tea1-tdT is mainly localized to one pole. A more careful analysis of Tea1 localization in rgf1-null cells, including wild-type cells as a control, is necessary. Additionally, the authors state that Tea1 is mainly localized to the non-growing end (lines 164), but this information is not shown in panel C or panel B.

Figure 1H. The authors suggest that the limited amount of Tea4-Tea1 at the growing pole underlies the curly phenotype of microtubules. Alternatively, it is possible that the curly microtubules and the extended dwelling time at the tips of rfg1-null cells contribute to the reduced amount of Tea1-Tea4 at the poles. In this context, it has been reported that Amo1 cooperates with Tea1 in maintaining cell polarity by regulating the proper dynamics of microtubules. Consequently, when Amo1 is absent, dwelling time at the tips is also lengthened, leading to curly microtubules and irregular Tea1 deposition (Pardo and Nurse, 2005). In addition, the double mutant amo1-null tea1-null forms branches during exponential growth, a phenotype not observed in the single mutants. These phenotypes are similar to those observed in the absence of Rgf1; therefore, microtubules may play a partial role or Rgf1 might be regulating Amo1.

Finally, it has been shown that an increase in the pool of free monomeric actin is sufficient to cause a switch from monopolar to bipolar actin in fission yeast cells, provided that both cell ends are marked as potential sites of growth (Rupes et al., 1999). Many monopolar mutants can induce NETO after this treatment, and therefore able to recognize both cell ends and to use them as sites for growth. It will be interesting to know whether this is the case for Rgf1.

2. Rgf1 cooperates with Mod5 in Tea4, anchoring to the cellular poles (Figure 2). According to the literature, in mod5-null cells, Tea1 remains associated with microtubule ends, but it no longer accumulates at both cell tips. The authors claim that rgf1-null cells show a similar defect (line 212), however, it remains to be precisely determined the impact of rgf1 deletion on Tea1 localization. From the data provided in Figure 2A, this can only be concluded for Tea4, and, in this case, signal can be appreciated at one of the poles. Authors should analyze the impact of Rgf1 and Mod5 deletion on Tea1 localization and include quantification of fluorescence enrichment at the cell tips for these experiments. Only then can they conclude that Rgf1 and Mod5 collaborate to position the polarity markers at the cell tips (lines 251-252)

Figure 2B presents an inconsistency between the results reported in this manuscript and those in Snaith and Nurse (2003). In the latter study, the percentage of branched cells in polarity re-establishment experiments in the presence of MBC (50 g ml-1) in mod5- cells reaches 84%, which is not statistically significantly different from tea1-null cells. In this manuscript, however, the authors indicate that only the rgf1-mod5- double mutant phenocopies tea1-null cells in the presence of MBC. The authors should clarify this discrepancy and include a statistical analysis of the results in Figure 2B.

Finally, it is recommended that the authors include TEA4-null cells in polarity re-establishment experiments and generate the double cdc11 tea4- mutant to determine the percentage of T-shaped cells in Figure 2B, C. This will provide a more comprehensive understanding of the phenotype of the rgf1- mod5- double mutant.

3. Rgf1 interacts with the cell-end marker Tea4 and binds to phosphatidylinositol-4-phosphate through its PH domain (Figure 3). From Figure S3, the authors conclude that Tea1 and Rgf1 do not interact. Nevertheless, it is recommended that the experiments be repeated under more stringent conditions, as an unspecific band appears that may mask a subtle interaction. This will ensure the validity of the findings and provide reliable insights into the relationship between Tea1 and Rgf1. In the pull-down assays a slight precipitation of Tea1 can be observed. Authors should demonstrate whether this is mediated by Tea4 binding.

In Figure 3F-H, the authors investigate the significance of the PH domain in the localization of Rgf1 at the poles. This analysis is not entirely novel, as previous research (Muñoz et al., 2014) characterized a version of Rgf1 lacking 56 amino acids in the PH domain and demonstrated that the normal localization of Rgf1 at the two tips was disrupted, with the signal primarily being monopolar. In that paper, the impact of the absence of the PH domain on the fluorescence intensity and protein level of Rgf1 was reported to be less severe than in the current study. The reasons for this discrepancy are not clarified by the authors. Furthermore, the role of phosphorylated phosphoinositides, specifically phosphatidylinositol 4,5-biphosphate (PIP2), in Rgf1 localization was demonstrated by Deng et al. (2005) in a JBC paper, which is not cited by the authors. In this work, the absence of PIP5K Its3 activity led to the severe disruption of Rgf1 localization at the cell tips.

4. Tea4 accumulation at the cell ends depends on Rgf1 anchoring to the PM and Rho1 activation (Figure 4).

Figure 4A shows that in Rgf1-null, rgf1-PH, rgf1-PTTR mutants, Tea4-GFP is equally distributed in many cells in both poles, whereas previously (Figure 1), it was claimed to accumulate at the non-growing end. To provide a more accurate analysis of this phenotype, the authors should measure the relative fluorescence intensity for all strains depicted at the non-growing and at the growing poles. Additionally, there may be an additive effect of both the lack of PH and the absence of the four amino acids critical in the GEF domain.

Figure 4B indicates that protein levels of rgf1-PH are severely reduced; therefore it is difficult to draw a clear conclusion from this type of assay.

In Figure 4E, the authors state that Rgf1 is required to maintain Tea4 at the poles, and when its activity is impaired, Tea4 levels are reduced, leading to a significant increase in T-shaped cells in the presence of MBC. Tea4-null cells should be included in this analysis, as previously suggested for Figure 2B. Also, it would be interesting to analyze whether artificial targeting of Tea4 to the poles in the absence of rgf1 restores the wild-type phenotype.

5. Rgf1 is part of actin-dependent machinery that signals growth poles in the absence of the Tea1-Tea4 complex (Figure 5). The authors demonstrate that Rgf1 control of cell polarity acts in parallel to the Tea1-Tea4 complex and suggest that it might involve actin, although the specific molecular mechanisms remain unclear. According to previous work by the authors, in the absence of Rgf1, actin cables are mostly normal and actin patches concentrate at the growing tip. These abnormalities do not exactly match the perturbations on the actin cytoskeleton imposed by stressors or during stationary phase. There might be some overinterpretation of the data.

Figure 5A. The description of these results is not clear. The authors indicate that stress treatments also induced branching in tea1-null mutant for 3 days in liquid medium, but it is not entirely clear if this is what is shown in panel A. In this panel, stationary cells are displayed, and it is noted that they have not been treated with stressors.

Figure 5G-I . LatA treatment disassembles both actin patches and cables, which contribute to endocytosis and transport, respectively. Specifying which actin structures are relevant in this control is important since both contribute to polarized growth, and this information is lacking in the manuscript. To complete this analysis, I will also include additional regulators of Cdc42, such as Scd1 and Scd2 (GEFs for Cdc42).

Rev. 2:

This manuscript examines cell polarity in the fission yeast S. pombe. Cell tip growth has been known to depend on the Tea1-Tea4 complex that is delivered to cell tips by dynamic microtubules. In this paper, the authors propose that Rho1-GTPase activity, under control of the GEF Rgf1 and the actin cytoskeleton, serve as a secondary signal for cell tip growth. A combination of microscopy and biochemistry demonstrate that Rgf1 interacts with Tea4 and is required for stable localization of Tea4 at cell tips. Combined loss of Rgf1 and Mod5, which also stabilizes Tea1-Tea4 at tips, leads to dramatic T-shaped cells. This activity of Rgf1 might depend on its membrane-binding domain, although these data are not strong (see below). The authors also define regulation of the Rgf1-Rho1 pathway during stress conditions, which have been known to impact the Tea1-Tea4 pathway and its mutant phenotypes. Overall, the paper represents a significant contribution for the field because it identifies an actin-based cell polarity pathway that functions in parallel with the previously defined microtubule-based cell polarity pathway. Most of the data are very strong, with several exceptions that are described below and can be fixed. The work will draw interest for researchers studying cell polarity and stress signaling in many cell types, and the themes of this work will likely apply broadly.

Specific comments:

1. The authors show that Tea4 localization at cell tips is unstable in rgf1∆ mutants, similar to mod5∆ and tea3∆ mutants. This result begs the question if Mod5 and Tea3 are properly localized in rgf1∆ mutants. It seems possible that Rgf1 acts on Tea4 through these other anchoring proteins, although the genetic data also support the possibility of parallel action by multiple anchors. Therefore, testing Mod5 and Tea3 localization in rgf1∆ cells is interesting either way.

2. The authors claim that Rgf1 dots and Tea4 dots colocalize, but the images in Figure 3A are not strong enough to support this conclusion. I would strongly suggest that the authors use their innovative "head-on" imaging (as in Figure 1F) to test if Rgf1-tdT and Tea4-GFP dots colocalize at cell tips.

3. Figure 3C: This western blot shows results for co-immunoprecipitation assays for Rgf1 interactions with Tea1 and Tea4. I am confused why/how Tea1-GFP and Tea4-GFP are shown at the same molecular weight in the gels. They have different molecular weights (Tea1 is 127 kDA, and Tea4 is 90 kDa), so they should migrate as differently sized bands by SDS-PAGE.

4. Figure 3D: For the yeast two-hybrid assays, the authors need to include the negative control of Tea4 and Tea1 alone (it would be -/Tea4 and -/Tea1 by their nomenclature). This would control for autoactivation by these constructs.

5. The authors make a rgf1∆PH mutant that lacks the lipid-binding domain and observe defects in this mutant. However, the mutant is expressed at massively reduced levels compared to wildtype (as shown in Figure 3H and mentioned in the text). The reduced expression makes it impossible to interpret any role of membrane binding, although the authors conclude that membrane-binding is important for cell tip localization and function. I would argue that protein expression is important for cell tip localization, since the low concentration will reduce the levels at cell tips. I would also argue that phenotypic defects for this mutant are due to its extremely low levels. It seems very important for the authors to test this construct under increased expression levels, so that its levels are similar to wildtype. They can perform this experiment with an exogenous promoter. Otherwise, I disagree that the mutant has compromised localization and function. Importantly, this problem extends to other parts of the paper as well because the authors use the same under-expressed mutant to test Rho1 activation, Tea4 binding, cell branching, and more. In my opinion, all of these experiments need to be repeated under increased expression of Rgf1∆PH.

6. Given the previous point, it would be helpful for the authors to validate that Rgf1-∆PTTR mutant is expressed as similar levels as wildtype Rgf1.

7. An alternative interpretation of the synthetic defects in rgf1∆ tea4∆ cells is that rgf1∆ induces cell stress. Most of the phenotypes examined for tea1∆ and tea4∆ cells are exacerbated by cell stress or by rgf1∆. The authors might consider this possibility in their Discussion section. It would be a less exciting model, but nonetheless seems plausible from the available data.

8. The authors propose feedback between Rgf1 and the actin cytoskeleton. This is an interesting idea, but I do not see evidence that Rgf1 regulates the actin cytoskeleton. Are there actin defects beyond monopolar growth in rgf1∆ mutants? Or

alternatively, does the actin-Rgf1 relationship act in one direction with actin upstream of Rgf1?

9. At the bottom of page 22, the authors state that Tea4 and Rgf1-Rho1 are essential for each other's cell tip localization. However, I don't recall data for Rgf1-GFP localization in tea4∆ cells. Can the authors clarify if there is a defect?

10. Minor: In line 164, the authors state "Tea1 also localized…" but I think they mean "Tea4 also localized…"

11. Minor: on line 267, there is a typo in Schizosaccharomyces.

12. Minor: Figure 5A should be relabeled to spell out "stationary phase" because "S-phase" has a different meaning in the cell cycle.

Academic Editor’s comments

This is an interesting manuscript by Sanchez and colleagues.

The first part of the manuscript supports a role for Rgf1 in the Tea1/4 pathway, by acting as an anchor of Tea4 at cell poles. This part of the manuscript is well supported, although it is unclear at present whether Rgf1 acts as physical anchor or acts indirectly (in unknown manner) through activation of Rho1. There are a few issues that would need to be addressed, listed below, but this part is not the main point of the paper.

A) The authors show that Tea4-GFP localization at cell poles is vastly reduced in rgf1∆ mod5∆ double mutants. It would be interesting to test the localization of Tea1-GFP in the rgf1∆ mod5∆ double mutants to check whether Rgf1 functions upstream of Tea1 or in parallel to it to localize Tea4.

B) The section on the role of Rgf1 activity and localization is a bit overstated. Because the ∆PH mutant is unstable, it is impossible to make strong claims about the role of Rgf1 localization. It could be that the phenotype of this mutant is due to loss of protein stability. Similarly, at the bottom of page 11, it is not possible to claim that “GEF activity is more critical than GEF localization”. The data show that GEF activity is critical, but no strong conclusion can be made on the importance of localization.

C) It is not clear to me how Rgf1 may function to localize Tea4 at cell poles yet be delocalized to lateral patches upon stresses (Figure 5E), while Tea1/4 isn’t. This makes it a bit unlikely that Rgf1 serves to mark the pole. This could be discussed

The main message of the paper is that Rgf1 forms a second pathway to define sites of growth in parallel to the Tea1/4 pathway delivered by microtubules. The main argument for this model is that double mutants of tea1∆ rgf1∆ form T-shaped cells even when not stressed by overgrowth, osmotic or heat treatments, conditions known to transiently depolarize the actin cytoskeleton and lead to formation of T-shaped cells in tea1/4∆ cells upon growth re-initiation. The authors interpret this finding as Rgf1 (acting through activation of Rho1) forming a second pole-marking pathway dependent on the actin cytoskeleton. I think it is equally possible (in my view quite likely) that the reason for the phenotype of rgf1∆ tea1∆ is that cells lacking Rgf1 experience transient loss of polarization, which acts is much the same way as other stresses that cause transient actin depolarization. This is a question that needs to be addressed, for instance by examining when tea1∆ rgf1∆ double mutant cells form Ts: Is it at random times? Or does growth stop because of low Rho1 activity (such as upon a CW failure that is successfully repaired) in rgf1∆ and therefore this acts as a stress? This will allow to establish whether rgf1∆ acts as a stressor, or whether Rgf1 functions in parallel to Tea1.

Beyond this issue on genetic interpretation of the double mutant phenotype, the Sawin group has recently shown that loss of F-actin does not lead per se to loss of polarity, nor polarized growth, but causes activation of the stress-activated MAPK pathway (SAPK; through the MAPK Sty1). This raises the question of whether all of the cases shown here to lead to T-shaped cells, and interpreted to act through the actin cytoskeleton, are not all due to activation of the SAPK (rather than F-actin depolymerization per se), including the rgf1∆ genotype. One simple experiment to start with is to probe whether branches form in tea1∆ rgf1∆ sty1∆ triple mutant. Further tools, for instance developed by the Sawin group, are available to test this hypothesis.

In summary, my view is that the paper has potential, but the main message is not (yet) well supported, and I would not be surprised if more detailed investigation reveals that rgf1∆ activates the SAPK, and that this causes the observed phenotypes.

---

## [Decision Letter · Decision Letter 2]

12 Sep 2024

Dear Dr Sanchez,

Thank you for your patience while we considered your revised manuscript entitled "Rgf1 GEF activity toward Rho1 defines a new actin-dependent signal to determine growth sites independently of microtubules and Tea1" for publication as a Research Article at PLOS Biology. This revised version of your manuscript has been evaluated by the PLOS Biology editors, the Academic Editor and two of the original reviewers.

Based on the reviews (attached below) and after discussions with the Academic Editor, we are likely to accept this manuscript for publication, provided you satisfactorily address some of the remaining points raised by Reviewer 1 as follows:

1. "inconsistency in the findings regarding the effect of artificial targeting of Tea4 to the cell pole"

This seems to refer to the CRIB-GFP Tea4-GBP experiment of Figure 4F and I agree that it is not a strong experiment, as the lack of effect could simply point to a not fully-functional Tea4 in these conditions. But if Rgf1 also functions in parallel to Tea4, as shown in the 2nd half of the manuscript, then lack of effect also makes sense. Please clarify this further in the text.

2. "actin distribution data in rgf1∆ cells" – this is acceptable as is.

3. "insufficient demonstration of the mechanism by which Rgf1 acts redundantly in establishing polarized growth"

I disagree. The data is now quite clear and well supported, and is novel. The authors could comment on possible link with the reported study on Bgs1. However, in that study, effect of double mutant is to cause complete loss of polarity (round cells), which is different from the rgs1∆ tea4∆ double mutant shown here that causes T-shaped cells (i.e. loss of polarity positioning). There is also nothing really shown in the Brunner Cell paper.

4. Statement that "that the activation of Rho1, rather than Cdc42, is responsible for defining the growth sites" – it seems the intention is to state this in the context of tea1/4∆ mutants. Perhaps this can be further clarified in the text, as it is indeed clear that Cdc42 plays important roles in WT cells.

Please also make sure to address the following data and other policy-related requests:

1. TITLE

We would like you to consider a suggestion to improve the title:

"Rho1 and Rgf1 establish a new actin-dependent signal to determine growth poles in yeast independently of microtubules and the Tea1-Tea4 complex"

2. DATA POLICY

Thank you for submitting the file containing all the data underlying the graphs shown in the figures. However, we have noted that in Fig. 3H; Fig. 4E, F; Fig. S5B and K, there are only two values represented in the bar graphs, and they have an error bar, which is misleading. Please remove the error bars and represent the two values as dots for each of the graphs.

In addition, please indicate in the corresponding figure legends in your manuscript where the underlying data can be found. For example, you could add at the end of each figure legend: "The data underlying the graphs shown in the figure can be found in S1 Data." Please also use that nomenclature for the file, rather than Data S1.

3. CODE POLICY

We expect to receive your revised manuscript within two weeks. 

*Published Peer Review History*

*Press*

Sincerely,

Ines

--

Ines Alvarez-Garcia, PhD

Senior Editor

PLOS Biology

Reviewers' comments

Rev. 1:

Thank you for the opportunity to assess the revised version of this manuscript. In my previous evaluation, I expressed concerns about the data presentation, experimental design, and quantification of several experiments, largely regarding the analysis of how Rgf1 contributes to the anchoring of Tea4 and Tea1 polarity markers at the pole. I am pleased to report that the authors have satisfactorily resolved most of these concerns in the revised manuscript. However, there are still some aspects that have been insufficiently developed in the current manuscript.

First, it is necessary to address the inconsistency in the findings regarding the effect of artificial targeting of Tea4 to the cell pole on the formation of T-shaped cells in the rgf1� mutant. The authors failed to provide an explanation for this discrepancy, and it is unclear why targeting Tea4 to the poles does not suppress the formation of T-shaped cells in this mutant. If the lack of anchoring of Tea4 at the poles results in a decrease in polarity markers and is the cause of the observed phenotype, it is expected that anchoring Tea4 to the poles would alleviate or prevent it. Second, I am not convinced of the representation of actin distribution data in rgf1� cells. Maybe the authors could use a more quantitative approach. In addition, the actin probe LifeAct-GFP, driven by a strong actin promoter, has been shown to affect growth and division. I advise the authors to repeat experiments using high-resolution confocal microscopy and alternative and less detrimental means such as phalloidin staining, as in their original work, to validate these data.

Nevertheless, my primary reservation about this work remains the insufficient demonstration of the mechanism by which Rgf1 acts redundantly in establishing polarized growth in fission yeast. My opinion on this aspect has remained unchanged. I convey that the main message of this study is that Rgf1 forms a second pathway dependent on the actin cytoskeleton to define sites of growth in parallel to the Tea1/4 pathway delivered by microtubules. However, I have concerns regarding the novelty of this observation and the extent to which this conclusion was developed in the current manuscript. That F-actin cooperate with Tea1-Tea4 complex in the establishment of polarity was already published in Cell by Damian Brunner lab in 2016 (10.1016/j.cell.2016.04.037 ). This study extends this observation by investigating Rgf1 as an activator of Rho1. More recently, a related study by Ribas et al. at the Instituto de Biología Funcional y Genómica (IBFG, Salamanca) was published in iScience (Ramos et al., iScience, 2024, Doi: 10.1016/j.isci.2024.110477). In this study, the authors propose a similar redundant pathway to Tea1-Tea4 in growth polarity establishment involving Bgs1 and actin patches. Both the absence of Bgs1 function and defective actin patch localization caused a similar total loss of growth polarity in the absence of the Tea1-Tea4 complex. This work suggests that Bgs1 is necessary for the correct localization of actin patches and sterol-rich domain domains, which, in turn, are ultimately responsible for the control of growth polarity along with the Tea1-Tea4 complex. This raises the question whether Bgs1 is one of the targets of Rho1.

Moreover, the authors did not provide sufficient evidence to explain the involvement of both actin patches and cables in the formation of T-shaped cells. This aspect has not been adequately addressed in this study. Furthermore, in the Discussion section, the authors posit that the activation of Rho1, rather than Cdc42, is responsible for defining the growth sites. However, this conclusion is based solely on the findings obtained using the gef1 mutant. The authors did not explore whether other GEFs, which are more relevant to Cdc42 activation during polarized growth, participate in this process. As a result, I believe that this assertion should be removed from the manuscript.

Rev. 2:

I am satisfied with the authors' revisions. They performed a number of new experiments to address reviewer comments, I was particularly impressed with increased expression of the PH mutant using the Pact1 promoter. This paper represents a nice contribution to the field and raises some interesting questions for the field to address in the future.

---

## [Editor Report · Decision Letter 3]

13 Oct 2024

Dear Dr Sanchez,

Thank you for the submission of your revised Research Article entitled "Rho1 and Rgf1 establish a new actin-dependent signal to determine growth poles in yeast independently of microtubules and the Tea1-Tea4 complex" for publication in PLOS Biology. On behalf of my colleagues and the Academic Editor, Sophie Martin, I am delighted to let you know that we can in principle accept your manuscript for publication, provided you address any remaining formatting and reporting issues. These will be detailed in an email you should receive within 2-3 business days from our colleagues in the journal operations team; no action is required from you until then. Please note that we will not be able to formally accept your manuscript and schedule it for publication until you have completed any requested changes.

PRESS

Sincerely, 

Ines

--

Ines Alvarez-Garcia, PhD

Senior Editor

PLOS Biology
